# Toward Bit-Efficient Dataset Condensation: A General Framework

## Abstract

Dataset condensation aims to distill a large-scale dataset into a compact set of synthetic samples for efficient training. Existing methods primarily focus on reducing the number of samples but generally assume full-precision representations. While effective, this assumption limits their applicability in resource-constrained scenarios due to several major drawbacks: (1) *Transmission bottlenecks*—full-precision datasets consume excessive bandwidth and introduce latency during network transfer, especially in cloud–edge collaborative learning; (2) *Memory overhead*—storing and processing full-precision data rapidly exhausts GPU memory or RAM, restricting batch sizes; and (3) *Hardware underutilization*—modern accelerators are optimized for low-precision operations, yet full-precision data prevents full efficiency gains in training and inference. To address these challenges, we propose a novel approach that fine-tunes distilled full-precision datasets into compact low-bit representations, substantially reducing memory usage with minimal computational overhead. Central to our method is a differentiable bit-conscious optimization framework. This framework allows more synthetic samples to be stored within the same memory budget, thereby improving downstream performance. Beyond the algorithmic contribution, we provide theoretical analysis that characterizes (1) the trade-off between compression error and generalization error under memory constraints, and (2) the extent to which Fisher information is preserved under bit compression. Extensive experiments compared to state-of-the-art baselines validate both the effectiveness and efficiency of our method.

## 1 Introduction

Dataset condensation is an emerging technique that synthesizes a compact dataset from a large-scale one, enabling more efficient training of machine learning models. Unlike traditional data selection or compression methods, condensation generates artificial samples that are explicitly optimized to preserve the key learning properties of the original data. As a result, models trained on these synthetic datasets can achieve comparable performance to those trained on the full dataset. This paradigm is particularly appealing in resource-constrained scenarios, such as deployment on edge devices, distributed learning environments, or continual learning settings where storage and communication overheads are critical bottlenecks.

Despite advances in dataset condensation (Zhao & Bilen, 2021; Zhou et al., 2022), most methods assume that synthetic samples are stored and used in full precision. While effective, this design limits their practicality in low-resource settings. Using low-bit representations addresses several major challenges: (1) *Transmission*: Large high-precision datasets can saturate networks, especially wireless links, causing bandwidth and latency issues. In contrast, low-bit data requires less bandwidth, enabling efficient transfer between cloud servers and edge devices in distributed or federated learning. (2) *Memory*: Data storage in GPU memory or RAM is often a bottleneck. Low-bit datasets reduce the memory footprint, allowing larger batches of data to fit in memory and improving training efficiency—crucial for devices with limited RAM. (3) *Hardware efficiency*: Many modern accelerators are optimized for low-precision operations. Storing distilled datasets in low-bit formats allows full use of these hardware capabilities, yielding faster training and inference.

In contrast, the benefits of bit compression—widely studied in the context of model compression and efficient inference—have not yet been explored for representing the synthetic datasets themselves.

Moreover, bit compression introduces non-differentiable operations, making it challenging to integrate into the gradient-based dataset condensation optimization frameworks that underpin most dataset condensation methods. To bridge this gap, we introduce a novel and efficient dataset condensation framework that *fine-tunes* full-precision distilled datasets to generate highly compact low-bit data representations with only minimal computational burden, significantly enhancing storage efficiency. Specifically, we introduce a differentiable condensation optimization algorithm tailored for reducing the number of bits to represent synthetic data. Our approach allows the condensation process to jointly optimize data content and its low-bit representation, enabling the generation of a substantially larger number of samples under the same memory budget. This flexibility is crucial in maximizing the utility of condensed data in memory-constrained applications. We name our approach as *Bit-Conscious Dataset Condensation* (BCDC).

In addition to our algorithmic contributions, we provide extensive theoretical analysis that characterizes: (1) the fundamental trade-off between the quantization error and generalization error under fixed memory constraints; and (2) Fisher information preservation under bit compression. These analyses offer deeper insights into how quantization affects the effectiveness of condensed datasets and guide practical design choices.

Comprehensive experiments across multiple benchmarks and state-of-the-art baselines show that: (1) our method delivers strong performance while significantly reducing memory usage when using the same number of distilled images; and (2) under a fixed memory budget, it accommodates more distilled images, resulting in markedly improved performance and advancing the practicality and scalability of dataset condensation.

Our main contributions in this paper are summarized as follows:

- We propose a novel and general dataset condensation framework through low-bit compression, enabling memory-efficient learning.
- We develop an efficient dataset condensation optimization algorithm, facilitating more effective dataset distillation.
- The theoretical analysis is derived to characterize: (1) the trade-off between quantization error and generalization error under memory constraints; and (2) Fisher information preservation under bit compression.
- Extensive experiments are conducted on multiple benchmarks and state-of-the-art baselines to validate the effectiveness and efficiency of the proposed method.

## 2 RELATED WORK

### 2.1 CORESET SELECTION AND DATA CONDENSATION

**Coreset Selection** Coreset selection (Har-Peled & Mazumdar, 2004) aims to identify a small, representative subset of the original dataset such that training a model on this subset yields performance comparable to training on the full dataset. Importantly, the selected subset consists of actual data samples rather than synthetic ones. This idea has been extensively explored in domains such as active learning (Settles, 2009) and continual learning (Lopez-Paz & Ranzato, 2017), where the goal is to select the most informative examples for training (Yang et al., 2023b; Welling, 2009; Chen et al., 2010; Rebuffi et al., 2017; Aljundi et al., 2019). However, coreset selection inherently relies on choosing a portion of the original dataset and thus may fail to capture useful information contained in the remaining samples.

**Data Condensation** Dataset condensation (DC) (Wang et al., 2018) synthesizes compact data that retains the essential information of the original dataset, enabling efficient model training with fewer samples. Unlike coreset selection, which chooses real data points, condensation generates synthetic ones informed by the full dataset. Approaches include: *(I) Bi-level Optimization* (e.g., DD (Wang et al., 2018), Qin et al. (Qin et al., 2024), EDC (Shao et al., 2024)); *(II) Analytical Methods* such as KRR (Nguyen et al., 2021); *(III) Surrogate Matching*, including gradient (DC (Zhao et al., 2021), DSA (Zhao & Bilen, 2021)), trajectory (MTT (Cazenavette et al., 2022)), and loss/feature matching (LCMat (Shin et al., 2023), CAFE (Wang et al., 2022a), DM (Zhao & Bilen, 2023)); *(IV) Parameter-Efficient Methods* using data partitioning (IDC (Kim et al., 2022), IDM (Zhao et al., 2023), DQ (Zhou

et al., 2023)), basis factorization (HaBa (Liu et al., 2022), RememberThePast (Deng & Russakovsky, 2022)), or low-rank techniques (LoDC (Yang et al., 2023a)); *(V) Regularization* (DWA (Du et al., 2024), CMI (Zhong et al., 2025)); *(VI) Diffusion Models* (D³HR (Zhao et al., 2025), D⁴M (Su et al., 2024)); and *(VII) Optimization-Free Methods* like RDED (Sun et al., 2024).

Despite these advances, most approaches rely on full-precision representations, limiting efficiency in resource-constrained settings. In contrast, our method targets low-bit synthetic data generation, significantly reducing memory usage while maintaining competitive model performance.

## 2.2 CONTINUAL LEARNING

Continual learning (CL) seeks to enable models to learn from non-stationary data distributions without forgetting knowledge acquired from previously encountered tasks. Most existing CL approaches rely on storing and replaying raw data samples (Kirkpatrick et al., 2017; Schwarz et al., 2018; Zenke et al., 2017; Rebuffi et al., 2017; Chaudhry et al., 2018; Lopez-Paz & Ranzato, 2017; Riemer et al., 2019; Chaudhry et al., 2019; Buzzega et al., 2020; Prabhu et al., 2020; Pham et al., 2021; Verwimp et al., 2021; Arani et al., 2022; Caccia et al., 2022; Wang et al., 2022c) or on using synthetic data generated at full precision (Yang et al., 2023a). However, data efficiency and privacy remain critical challenges in CL, as raw data from previous tasks may be unavailable or sensitive during the training of new tasks. In this paper, we integrate our BCDC into CL, enabling the training of CL models that simultaneously improve data efficiency, enhance privacy, and maintain strong performance.

## 2.3 LOW BITS QUANTIZATION

To our best knowledge, BCDC is the first to explore bit-efficient dataset distillation. Unlike network quantization (Wang et al., 2022b; Gong et al., 2019; Yao et al., 2021), which targets model parameters, our method focuses on reducing the memory footprint of the distilled dataset itself. Moreover, our approach is seamlessly compatible with existing dataset distillation pipelines.

There is no prior work that integrates quantization into dataset condensation. Existing DC methods always operate with full-precision samples, and quantization research focuses exclusively on model parameters or activations rather than synthetic data. This gap directly motivates BCDC, which is the first framework to introduce quantization-aware dataset condensation.

# 3 METHOD

## 3.1 PROBLEM DEFINITION

**Traditional Dataset Distillation** The goal of dataset distillation is to distill a large-scale dataset of $\mathcal{T} = \{(\boldsymbol{x}_i, y_i)\}_{i=1}^{i=N}$ into a small-scale dataset $\mathcal{S} = \{(\boldsymbol{x}_i, y_i)\}_{i=1}^{i=m}$, where $m \ll N$. The objective is to ensure that the network trained on the compressed dataset, with parameters $\boldsymbol{\theta}^{\mathcal{S}}$, achieves performance comparable to that of the network trained on the original dataset $\mathcal{T}$, with parameters $\boldsymbol{\theta}^{\mathcal{T}}$, where:

$$\boldsymbol{\theta}^{\mathcal{T}} = \arg\min_{\boldsymbol{\theta}^{\mathcal{T}}}[\mathcal{L}(\boldsymbol{\theta}^{\mathcal{T}}, \mathcal{T}) = \frac{1}{|\mathcal{T}|} \sum_{(\boldsymbol{x}, y) \sim \mathcal{T}} \mathcal{L}(\boldsymbol{x}, y, \boldsymbol{\theta}^{\mathcal{T}})] \qquad (1)$$

$$\boldsymbol{\theta}^{\mathcal{S}} = \arg\min_{\boldsymbol{\theta}^{\mathcal{S}}}[\mathcal{L}(\boldsymbol{\theta}^{\mathcal{S}}, \mathcal{S}) = \frac{1}{|\mathcal{S}|} \sum_{(\boldsymbol{x}, y) \sim \mathcal{S}} \mathcal{L}(\boldsymbol{x}, y, \boldsymbol{\theta}^{\mathcal{S}})] \qquad (2)$$

The dataset distillation can be formulated as a bi-level optimization problem:

$$\mathcal{S} = \arg\min_{\mathcal{S}} \mathcal{L}(\boldsymbol{\theta}^{\mathcal{S}}, \mathcal{T}) \quad \text{satisfy} \quad \boldsymbol{\theta}^{\mathcal{S}} = \arg\min_{\boldsymbol{\theta}^{\mathcal{S}}} \mathcal{L}(\boldsymbol{\theta}^{\mathcal{S}}, \mathcal{S}) \qquad (3)$$

where the inner loop optimizes a network to train on the synthetic dataset $\mathcal{S}$ and the outer loop optimization optimizes on the original dataset to learn the synthetic dataset.

**Low-Bits Dataset Quantization** Given an original dataset $\mathcal{T} = \{(\boldsymbol{x}_i, y_i)\}_{i=1}^{N}$, we first compress it into a small-scale full-precision dataset $\mathcal{S} = \{(\boldsymbol{x}_i, y_i)\}_{i=1}^{m}$, where $m \ll N$. Our goal is to further *fine-tune* a condensed low-bit dataset $\mathcal{S}_{\text{quant}} = \{(\tilde{\boldsymbol{x}}_i, y_i)\}_{i=1}^{m}$, where each $\tilde{\boldsymbol{x}}_i$ is a quantized version of $\boldsymbol{x}_i$ represented using $b$-bit precision.

## 3.2 PROPOSED METHOD

We adopt a uniform quantization method:

$$\boldsymbol{x}_{\text{low}} = \min \boldsymbol{x}, \quad \boldsymbol{x}_{\text{high}} = \max \boldsymbol{x}, \quad \Delta = \frac{\boldsymbol{x}_{\text{high}} - \boldsymbol{x}_{\text{low}}}{2^b - 1} \tag{4}$$

where $b$ denotes the number of bits used to encode each dimension of a data sample. $\Delta$ denotes the length of the quantization interval. The uniform quantization-dequantization can be defined as:

$$Q_b(\boldsymbol{x}) = \text{round}(\frac{\boldsymbol{x} - \boldsymbol{x}_{\text{low}}}{\Delta})\Delta \tag{5}$$

**Differentiable Data Quantization (DDQ)** The near-zero gradients of the uniform quantization function (Eq. 5) at most input values hinder effective training on the quantized data, resulting in unstable learning dynamics. To address this issue, we introduce a differentiable asymptotic function that approximates a uniform data quantizer. Specifically, DDQ replaces hard quantization with a smooth, continuous approximation.

$$\phi(\boldsymbol{x}) = s \cdot \tanh\big(k(\boldsymbol{x} - \boldsymbol{m}_i)\big), \quad \text{if } \boldsymbol{x} \in P_i = [\boldsymbol{x}_{low} + i\Delta, \boldsymbol{x}_{low} + (i+1)\Delta]$$

$$\text{where} \quad \boldsymbol{m}_i = \boldsymbol{x}_{low} + (i + 0.5)\Delta, \quad s = \frac{1}{\tanh(0.5k\Delta)}. \tag{6}$$

Where in Eq. 6, $s$ ensures that the outputs of $\phi(\boldsymbol{x})$ are normalized to $-1$ and $+1$ at the boundaries of quantization intervals. $\boldsymbol{m}_i$ denotes the midpoint of each quantization interval. We then define a soft quantization function (Eq. 7) to provide a smooth and differentiable approximation to Eq. (5):

$$Q_V(\boldsymbol{x}) = \boldsymbol{x}_{low} + (i + \frac{\phi(\boldsymbol{x}) + 1}{2})\Delta \tag{7}$$

**Bi-Level Optimization for Low-Bit Condensation** We optimize $\mathcal{S}$ in full precision but penalize deviations from quantized values:

$$\min_{\mathcal{S}} \underbrace{\mathbb{E}_{(\boldsymbol{x},y)\sim\mathcal{T}_{\text{val}}}[\mathcal{L}(f_{\boldsymbol{\theta}^*}(\boldsymbol{x}),y)]}_{\text{Validation loss}} + \lambda \underbrace{\|\mathcal{S} - Q_V(\mathcal{S})\|_2^2}_{\text{Quantization loss}}$$

$$\text{s.t.} \quad \boldsymbol{\theta}^* = \arg\min_{\boldsymbol{\theta}} \mathbb{E}_{(\tilde{\boldsymbol{x}},y)\sim Q_V(\mathcal{S})}[\mathcal{L}(f_{\boldsymbol{\theta}}(\tilde{\boldsymbol{x}}),y)]$$

*Inner Loop (Model Training)*: Update $\boldsymbol{\theta}$ on $Q_V(\mathcal{S})$ via SGD:

$$\boldsymbol{\theta} \leftarrow \boldsymbol{\theta} - \eta_1 \nabla_{\boldsymbol{\theta}} \mathcal{L}(f_{\boldsymbol{\theta}}(Q_V(\mathcal{S})), \mathbf{y})$$

*Outer Loop (Dataset Update)*: Compute $\nabla_{\mathcal{S}}\mathcal{L}_{\text{val}}$ via chain rule:

$$\nabla_{\mathcal{S}}\mathcal{L}_{\text{val}} = \frac{\partial\mathcal{L}_{\text{val}}}{\partial\mathcal{S}} + \frac{\partial\mathcal{L}_{\text{val}}}{\partial\boldsymbol{\theta}^*} \cdot \frac{\partial\boldsymbol{\theta}^*}{\partial Q_V(\mathcal{S})} \cdot \frac{\partial Q_V(\mathcal{S})}{\partial\mathcal{S}} \tag{8}$$

where

$$\frac{\partial\boldsymbol{\theta}^*}{\partial Q_V(\mathcal{S})} = -\left(\frac{\partial^2 L_{\text{train}}}{\partial\boldsymbol{\theta}^2}\right)^{-1} \frac{\partial^2 L_{\text{train}}}{\partial Q_V(S)\partial\boldsymbol{\theta}} \tag{9}$$

Derivations of Eq. 9 are presented in Appendix A. Then, we update $\mathcal{S}$ with the following gradients:

$$\mathcal{S} \leftarrow \mathcal{S} - \eta_2\left(\nabla_{\mathcal{S}}\mathcal{L}_{\text{val}} + \lambda\nabla_{\mathcal{S}}\|\mathcal{S} - Q_V(\mathcal{S})\|_2^2\right)$$

**Final Quantization** After convergence, we apply the following quantization to each image:

$$\tilde{\boldsymbol{x}}_i = \text{round}\left(\frac{\text{clip}(\boldsymbol{x}_i, \boldsymbol{x}_{\text{low}}, \boldsymbol{x}_{\text{high}}) - \boldsymbol{x}_{\text{low}}}{\Delta}\right)$$

### 3.3 Integrate BCDC with Existing Approaches

**Bi-level Dataset Condensation with Quantized Data Representations**  Our proposed BCDC serves as a general, versatile framework that integrates seamlessly with existing approaches. The proposed algorithm integrated with bi-level dataset condensation loss function (Wang et al., 2018) is presented in Algorithm 1. Integrating BCDC with surrogate loss function, e.g., DM (Zhao & Bilen, 2023) is presented in Algorithm 2 in Appendix.

---

**Algorithm 1** Bi-level Dataset Condensation with Quantized Data Representations

---

**Input:** Training set $\mathcal{T}$, distilled dataset learning rates $\gamma$
**Initialize:** Initialize distilled dataset and labels $\mathcal{S}$.
**while** not converged **do**
  Sample a training batch from the training set: $\{\boldsymbol{x}, y\} \sim \mathcal{T}$
  Perform $K$ optimization steps on inner objective to obtain $\boldsymbol{\theta}_i^*$
  Compute synthetic data gradient $\nabla_{\mathcal{S}} \mathcal{L}_{\text{val}}$ by Eq. (8)
  Update the distilled dataset: $\mathcal{S} \leftarrow \mathcal{S} - \gamma \nabla_{\mathcal{S}} \mathcal{L}_{\text{val}}$
  Train the model $\boldsymbol{\theta}_i$ on the current distilled dataset $\mathcal{S}$ for one step
**end while**

---

## 4 Theoretical Analysis

We use $\mathcal{D}$ to represent the *ground truth* distribution governing data generation, $R_{\mathcal{D}}(f_{\boldsymbol{\theta}}) = \mathbb{E}_{(\boldsymbol{x},y) \sim \mathcal{D}} \mathcal{L}(f_{\boldsymbol{\theta}}(\boldsymbol{x}), y)$, which is the expected loss (or risk) over the true data distribution and also known as generalization error. $\hat{R}_{Q_b(\mathcal{S})}(f_{\boldsymbol{\theta}}) = \mathbb{E}_{(\boldsymbol{x},y) \sim Q_b(\mathcal{S})} \mathcal{L}(f_{\boldsymbol{\theta}}(\boldsymbol{x}), y)$ denotes the empirical risk on the quantized data.

**Assumption 4.1.** A hypothesis function $f_{\boldsymbol{\theta}} : \mathbb{R}^d \to \mathbb{R}$ is called Lipschitz continuous with constant $L > 0$ if
$$\|f_{\boldsymbol{\theta}}(\boldsymbol{x}_1) - f_{\boldsymbol{\theta}}(\boldsymbol{x}_2)\| \le L \|\boldsymbol{x}_1 - \boldsymbol{x}_2\|, \quad \forall \boldsymbol{x}_1, \boldsymbol{x}_2 \in \mathbb{R}^d.$$
If $h$ is differentiable, this implies a bound on its gradient: $\|\nabla f_{\boldsymbol{\theta}}(\boldsymbol{x})\| \le L, \forall \boldsymbol{x} \in \mathbb{R}^d$.

**Assumption 4.2.** The empirical risk function $R_{\mathcal{D}}(f_{\boldsymbol{\theta}}) : \mathbb{R}^d \to \mathbb{R}$ is $\beta$-smooth if
$$R_{\mathcal{S}}(f_{\boldsymbol{\theta}}) \le R_{\mathcal{D}}(f_{\boldsymbol{\theta}}) + \langle \nabla R_{\mathcal{D}}(f_{\boldsymbol{\theta}}), f_{\boldsymbol{\theta}}(\mathcal{S}) - f_{\boldsymbol{\theta}}(\mathcal{D}) \rangle + \frac{\beta}{2} \|f_{\boldsymbol{\theta}}(\mathcal{S}) - f_{\boldsymbol{\theta}}(\mathcal{D})\|^2,$$

**Assumption 4.3.** $Q_b$ be a $b$-bit quantizer with $\mathbb{E}[\|Q_b(\boldsymbol{x}) - \boldsymbol{x}\|] \le C 2^{-b}$ (following (Gray & Neuhoff, 2002)) where $C$ is a constant.

**Assumption 4.4.** $\mathcal{H}$ be a hypothesis class with Rademacher complexity (Bartlett & Mendelson, 2002) $\mathfrak{R}_n(\mathcal{H}) \le \kappa/\sqrt{n}$, where $\kappa$ is a constant and $n$ denotes the number of training samples.

Let $M$ denote the total memory budget in bits and $d$ be the data dimension. $m(b) = \lfloor M/(bd) \rfloor$ denotes the number of stored samples under $b$-bit quantization.

**Theorem 4.5** (Memory-Constrained Quantization-Generalization Trade-off). *With probability* $\ge 1 - \delta$:
$$R_{\mathcal{D}}(f_{\boldsymbol{\theta}}) \le \underbrace{\hat{R}_{Q_b(\mathcal{S})}(f_{\boldsymbol{\theta}})}_{\text{Empirical Risk}} + \underbrace{\frac{2\kappa}{\sqrt{m(b)}} + \sqrt{\frac{\log(2/\delta)}{2m(b)}}}_{\text{Generalization Error}} + \underbrace{LC 2^{-b} + \beta C^2 2^{-2b}}_{\text{Quantization Error}} \quad (10)$$

**Implications:** Each term in the above generalization bound can be interpreted as the following: (1) The term $\frac{2\kappa}{\sqrt{m(b)}}$ reflects the reduced model complexity from more samples; (2) $\sqrt{\frac{\log(2/\delta)}{2m(b)}}$ is the classical Hoeffding concentration term; (3) $LC 2^{-b}$ shows the first-order quantization error; (4) $\beta C^2 2^{-2b}$ captures the second-order quantization effects.

The generalization bound in equation 10 exhibits a trade-off between quantization error and generalization error influenced by the choice of $b$, as shown in Table 1: selecting a smaller $b$ tightens the generalization bound but at the cost of increased quantization error; selecting a larger $b$ reduces quantization error but may lead to looser generalization bounds due to fewer samples. Balancing these factors is crucial for optimizing model performance under memory constraints.

Table 1: Trade-off between generalization error and quantization error.

| | $m(b)$ | $\frac{2\kappa}{\sqrt{m(b)}} + \sqrt{\frac{\log(2/\delta)}{2m(b)}}$ | $LC2^{-b} + \beta C^2 2^{-2b}$ |
|---|---|---|---|
| $b$ decreases ($\downarrow$) | $\uparrow$ | $\downarrow$ | $\uparrow$ |
| $b$ increases ($\uparrow$) | $\downarrow$ | $\uparrow$ | $\downarrow$ |

**Theorem 4.6** (Fisher Information Retention in Bit-Conscious Condensation). *Let $I(\mathcal{T}; \theta)$ and $I(Q_b(\mathcal{S}); \theta)$ denote the Fisher information of the original and quantized condensed datasets, respectively. For a b-bit quantizer:*

$$I(\mathcal{T}; \boldsymbol{\theta}) - I(Q_b(\mathcal{S}); \boldsymbol{\theta}) \leq \frac{L^2 \Delta^2}{8} tr \left( \mathbb{E} \left[ \nabla_{\boldsymbol{\theta}} \log p(\boldsymbol{\theta}|\mathcal{S}) \nabla_{\boldsymbol{\theta}} \log p(\boldsymbol{\theta}|\mathcal{S})^\top \right] \right)$$

*where $\Delta = 2^{-b+1}(\max(\mathcal{S}) - \min(\mathcal{S}))$. tr denotes the trace of the Fisher information matrix $\left( \mathbb{E} \left[ \nabla_{\theta} \log p(\theta|\mathcal{S}) \nabla_{\theta} \log p(\theta|\mathcal{S})^\top \right] \right)$.*

**Implications:** As $b$ increases, $\Delta_1$ decreases and the right-hand side bound becomes tighter, since more bits allow finer-grained image details to be preserved. Conversely, as $b$ decreases, $\Delta_1$ increases and the bound becomes looser, due to the loss of fine-grained detail with fewer bits. Due to space limitations, we provide detailed theorem proof in Appendix B.

## 5 EXPERIMENT

### 5.1 DATASET CONDENSATION FOR DEEP LEARNING

**Datasets** We assess the effectiveness of BCDC on the following benchmark datasets: MNIST (LeCun et al., 1998), CIFAR10 (Krizhevsky et al., 2009), CIFAR100 (Krizhevsky et al., 2009), TinyImageNet (Le & Yang, 2015) and ImageNet-1K (Deng et al., 2009).

**Baselines** We compare to both coreset selection and dataset distillation methods.

(I) *Coreset Selection:* Selecting a representative subset of real data: (1) *Random:* Selects images randomly from the dataset; (2) *Herding:* Selects samples heuristically, aiming for those closest to the class center (Welling, 2009; Belouadah & Popescu, 2020); (3) *Forgetting:* Selects samples that are most likely to be forgotten during model training (Toneva et al., 2019).

(II) *Dataset Distillation: DD* (Wang et al., 2018), *LD* (Bohdal et al., 2020), *DC* (Zhao et al., 2021), *DSA* (Zhao & Bilen, 2021), *MTT* (Cazenavette et al., 2022), *IDC* (Kim et al., 2022), *HaBa* (Liu et al., 2022), *RememberThePast* (Deng & Russakovsky, 2022) *DM* (Zhao & Bilen, 2023), *DataDAM* (Sajedi et al., 2023), TESLA (Cui et al., 2023), SRe$^2$L (Yin et al., 2023), *DWA* (Du et al., 2024), *RDED* (Sun et al., 2024), $D^4M$ (Su et al., 2024), *CMI* (Zhong et al., 2025), $D^3HR$ (Zhao et al., 2025).

**Implementation Details** Following the experimental protocol established in (Kim et al., 2022), we ensure that all methods operate under an equal memory budget. For each trial, we either select a coreset (Random, Herding, or Forgetting) or optimize a synthetic dataset (DD, LD, DC, DSA, DM, etc), and then use it to train 20 independently initialized ConvNet models (Rocco et al., 2017). All other hyperparameters are aligned with those used in prior work (Zhao et al., 2021; Zhao & Bilen, 2021; 2023). Each experiment setup is repeated five times, and we report the average test accuracy across runs. In addition, as detailed in Sec. 5.2, we assess the generalization capability of the synthetic datasets across architectures by evaluating them on five commonly used deep networks: ConvNet (Rocco et al., 2017), LeNet (LeCun et al., 1998), AlexNet (Krizhevsky et al., 2017), VGG11 (Simonyan & Zisserman, 2015), and ResNet18 (He et al., 2016). For this set of experiments, we use 2, 4, 4 and 4-bit representations for MNIST, CIFAR10, CIFAR100 and TinyImageNet respectively. The number of bits for each dataset is selected from $\{2, 4, 8\}$, based on the configuration that yields the highest validation performance. $\lambda = 0.2$. All experiments are conducted on a single NVIDIA A6000 GPU.

**Results and Analysis** Table 2 and 4 compares dataset condensation with coreset selection approaches, showing that dataset condensation generally outperforms coreset selection. In Table 3, we compare our BCDC with conventional dataset condensation methods (DC, DSA, DM) under two scenarios: using the same number of images (SI) and using the same memory budget (SM). The results highlight

Table 2: Comparison with coreset selection methods and dataset condensation methods.

| DataSet | Img/Cls | Coreset Selection Methods | | | Dataset Condensation Methods | | | | | |
|---|---|---|---|---|---|---|---|---|---|---|
| | | Random | Herding | Forgetting | DD | LD | DC | DSA | DM | DM+BCDC (Ours) |
| MNIST | 1 | 64.9±3.5 | 89.2±1.6 | 35.5±5.6 | - | 60.9±3.2 | **91.7±0.5** | 88.7±0.6 | 89.7±0.6 | **91.5±0.3** |
| | 10 | 95.1±0.9 | 93.7±0.3 | 68.1±3.3 | 79.5±8.1 | 87.3±0.7 | 97.4±0.2 | 97.1±0.1 | 96.5±0.2 | **97.8±0.3** |
| | 50 | 97.9±0.2 | 94.8±0.2 | 88.2±1.2 | - | 93.3±0.3 | 98.8±0.2 | **99.2±0.1** | 97.5±0.5 | 98.4±0.2 |
| CIFAR10 | 1 | 14.4±2.0 | 21.5±1.2 | 13.5±1.2 | - | 25.7±0.7 | 28.3±0.5 | 28.8±0.7 | 26.0±0.8 | **45.1±0.8** |
| | 10 | 26.0±1.2 | 31.6±0.7 | 23.3±1.0 | 36.8±1.2 | 38.3±0.4 | 44.9±0.5 | 51.1±0.5 | 48.9±0.6 | **60.6±0.5** |
| | 50 | 43.4±1.0 | 40.4±0.6 | 23.3±1.1 | - | 42.5±0.4 | 53.9±0.5 | 60.6±0.5 | 63.0±0.4 | **65.9±0.2** |
| CIFAR100 | 1 | 4.2±0.3 | 8.4±0.3 | 4.5±0.2 | - | 11.5±0.4 | 12.8±0.3 | 13.9±0.3 | 11.4±0.3 | **26.3±0.5** |
| | 10 | 14.6±0.5 | 17.3±0.3 | 15.1±0.3 | - | - | 25.2±0.3 | 32.3±0.3 | 29.7±0.3 | **38.3±0.7** |
| TinyImageNet | 1 | 1.4±0.1 | 2.8±0.2 | 1.6±0.1 | - | - | 4.61±0.2 | 4.79±0.2 | 3.9±0.2 | **10.6±0.4** |
| | 10 | 5.0±0.2 | 6.3±0.2 | 5.1±0.2 | - | - | 11.6±0.3 | 14.7±0.2 | 12.9±0.4 | **18.9±0.5** |

Table 3: Comparison with dataset distillation methods on the same number of image (SI) or same memory (SM).

| DataSet | Img/Cls | DC | +BCDC (SI) | +BCDC (SM) | DSA | +BCDC (SI) | +BCDC (SM) | DM | +BCDC (SI) | +BCDC (SM) |
|---|---|---|---|---|---|---|---|---|---|---|
| MNIST | 1 | 91.7±0.5 | - | **93.5±0.4** | 88.7±0.6 | - | **90.3±0.7** | 89.7±0.6 | 88.1±0.8 | **91.5±0.3** |
| | 10 | 97.4±0.2 | 96.0±0.2 | **97.7±0.4** | 97.1±0.1 | 96.2±0.3 | **97.8±0.3** | 96.5±0.2 | 95.7±0.6 | **97.8±0.2** |
| | 50 | 98.6±0.3 | 97.8±0.3 | **98.9±0.5** | 98.0±0.2 | 97.1±0.4 | **98.6±0.4** | 97.2±0.3 | 96.6±0.5 | **98.9±0.3** |
| CIFAR10 | 1 | 28.3±0.5 | 28.1±0.4 | **36.1±0.6** | 28.8±0.7 | 28.6±0.7 | **42.2±0.3** | 26.0±0.8 | 25.0±0.4 | **45.1±0.8** |
| | 10 | 44.9±0.5 | 43.1±0.5 | **51.2±0.4** | 51.1±0.5 | 49.7±0.6 | **57.3±0.4** | 48.9±0.6 | 47.1±0.8 | **60.6±0.5** |
| | 50 | 53.9±0.5 | 52.8±0.6 | **60.5±0.5** | 60.6±0.5 | 58.6±0.5 | **68.2±0.3** | 63.0±0.4 | 58.2±0.7 | **65.9±0.2** |
| CIFAR100 | 1 | 12.8±0.3 | 12.5±0.4 | **19.1±0.2** | 13.9±0.3 | 13.7±0.3 | **23.7±0.4** | 11.4±0.3 | 10.5±0.6 | **26.3±0.5** |
| | 10 | 25.2±0.3 | 24.6±0.3 | **27.6±0.5** | 32.3±0.3 | 30.9±0.2 | **34.2±0.6** | 29.7±0.3 | 28.6±0.4 | **38.3±0.7** |
| TinyImageNet | 1 | 4.61±0.2 | 4.32±0.3 | **7.10±0.2** | 4.79±0.2 | 4.52±0.4 | **10.80±0.5** | 3.9±0.2 | 3.7±0.3 | **10.6±0.4** |
| | 10 | 11.6±0.3 | 10.2±0.3 | **18.31±0.3** | 14.7±0.2 | 13.61±0.5 | **20.70±0.6** | 12.9±0.4 | 11.6±0.4 | **18.9±0.5** |

the following observations: (i) With an equal number of images, our method achieves more reductions in storage requirements—while maintaining performance close to that of the traditional DC approach. (ii) When the memory budget is the same, our method is able to store more samples within the same budget, thereby retaining richer information from the original dataset. As a result, it significantly outperforms existing condensation methods. For example, on CIFAR10 with 1 image per class, BCDC achieves improvements of 13.4% and 19.1% over DSA and DM, respectively. In Table 5, we compare our method against recent approaches—MTT (Cazenavette et al., 2022), DataDAM (Sajedi et al., 2023), TESLA (Cui et al., 2023), SRe²L (Yin et al., 2023) and DWA (Du et al., 2024)—on both Tiny-ImageNet and ImageNet-1K benchmarks. The results show that our BCDC can enhance dataset distillation performance under the same memory budget.

## 5.2 ABLATION STUDY

**Cross-Architecture Transferability Analysis** To assess how well BCDC generalizes across different model architectures, we perform a cross-architecture evaluation. Specifically, we generate the condensed dataset using a single architecture (such as AlexNet or ConvNet) and then use it to train five different network architectures from scratch. We evaluate the resulting models on the CIFAR-10 test set. As shown in Table 7, BCDC consistently delivers strong performance across these diverse architectures, highlighting its effectiveness in supporting cross-architecture knowledge transfer.

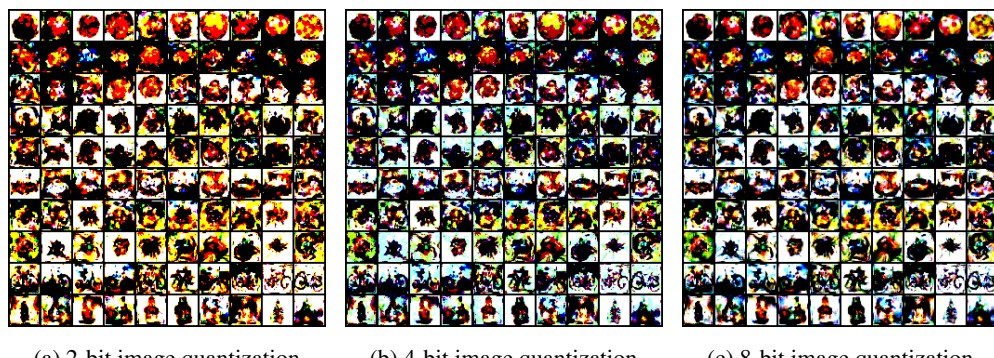

(a) 2-bit image quantization     (b) 4-bit image quantization     (c) 8-bit image quantization

Figure 1: Visualization of different bits image quantization by integrating BCDC with DC.

**Computation Efficiency Evaluation** To evaluate the training efficiency of our proposed BCDC compared to traditional dataset condensation methods without quantization, we report the training

Table 4: Compare with other advanced dataset condensation methods.

| | MTT | IDC-I | IDC | HaBa | RememberThePast |
|---|---|---|---|---|---|
| CIFAR10 (Img/Cls=1) | 46.3% | 36.7% | 50.6% | 48.3% | 66.4% |
| | MTT+BCDC | IDC-I+BCDC | IDC+BCDC | HaBa+BCDC | RememberThePast+BCDC |
| | 59.0% | 50.1% | 56.7% | 65.3% | 68.4% |
| CIFAR100 (Img/Cls=1) | MTT | IDC-I | IDC | HaBa | RememberThePast |
| | 24.3% | 16.6% | 24.9% | 33.4% | - |
| | MTT+BCDC | IDC-I+BCDC | IDC+BCDC | HaBa+BCDC | RememberThePast+BCDC |
| | 31.2% | 27.6% | 33.4% | 36.5% | - |

Table 5: Evaluation against state-of-the-art dataset distillation methods on Tiny-ImageNet and ImageNet-1K. Unless stated otherwise, we adopt the same model architecture during both the distillation and evaluation phases. Consistent with the configurations reported in their respective works, MTT (Cazenavette et al., 2022) and TESLA (Cui et al., 2023) employ ConvNet-128. In contrast, SRe$^2$L (Yin et al., 2023) generates synthetic data using ResNet-18 and assesses performance across ResNet-18, ResNet-50, and ResNet-101. The symbol † denotes that MTT is applied to a 10-class subset of the complete ImageNet-1K dataset.

| Dataset | ipc | ConvNet | | | ResNet-18 | | | ResNet-50 | | | ResNet-101 | | |
|---|---|---|---|---|---|---|---|---|---|---|---|---|---|
| | | MTT | DataDAM | TESLA | SRe$^2$L | DWA | +BCDC | SRe$^2$L | DWA | +BCDC | SRe$^2$L | DWA | +BCDC |
| Tiny-ImageNet | 50 | 28.0±0.3 | 28.7±0.3 | - | 41.1±0.4 | 52.8±0.2 | **55.1±0.3** | 42.2±0.5 | 53.7±0.2 | **55.6±0.3** | 42.5±0.2 | 54.7±0.3 | **57.8±0.4** |
| | 100 | - | - | - | 49.7±0.3 | 56.0±0.2 | **59.6±0.3** | 51.2±0.4 | 56.9±0.4 | **59.1±0.5** | 51.5±0.3 | 57.4±0.3 | **59.6±0.4** |
| ImageNet-1K | 10 | 64.0±1.3† | 6.3±0.0 | 17.8±1.3 | 21.3±0.6 | 37.9±0.2 | **39.6±0.3** | 28.4±0.1 | 43.0±0.5 | **45.5±0.6** | 30.9±0.1 | 46.9±0.4 | **49.2±0.3** |
| | 50 | - | - | 27.9±1.2 | 46.8±0.2 | 55.2±0.2 | **57.5±0.3** | 55.6±0.3 | 62.3±0.1 | **65.6±0.2** | 60.8±0.5 | 63.3±0.7 | **64.5±0.8** |
| | 100 | - | - | - | 52.8±0.3 | 59.2±0.3 | **61.7±0.4** | 61.0±0.4 | 65.7±0.4 | **67.8±0.3** | 62.8±0.2 | 66.7±0.2 | **68.0±0.4** |

cost comparison in Table 9 in Appendix D. Although BCDC introduces additional quantization-aware fine-tuning cost, it results in only a modest increase in training cost—ranging from 17% to 20%—while offering improved performance under quantized settings.

Table 8: Comparison of testing performance using naive quantization versus our BCDC.

| Method | Without Quantization | + Naive Quantization (Naive-Q) | + BCDC (Ours) |
|---|---|---|---|
| DC | 28.3±0.5 | 31.8±0.5 | 36.1±0.6 |
| DSA | 28.8±0.7 | 35.6±0.7 | 42.2±0.3 |
| DM | 26.0±0.8 | 34.3±0.2 | 45.1±0.8 |

**Effect of Bit-Width** $b$: Table 10 (in Appendix D) illustrates the effect of bit-width on dataset distillation performance. Lower bit-widths reduce the memory required per image, enabling the storage of a larger number of samples within a fixed memory budget. However, this reduction in precision also degrades image quality, which can hinder model performance. Conversely, higher bit-widths preserve more visual detail and improve image fidelity but increase the memory footprint per image, limiting the number of samples that can be stored. Empirically, we observe that performance improves as the bit-width increases from 2 bits, reaching its peak at 4 bits. Beyond this point, however, performance begins to decline despite the improved image quality, primarily due to the reduced number of stored samples. This highlights a trade-off between image quality and sample quantity, emphasizing the need to select an optimal bit-width that balances memory constraints and model performance in practical applications.

**Comparison Between Our BCDC and Direct Quantization:** To assess the effectiveness of our quantization-aware fine-tuning strategy, we compare the performance of our BCDC against a baseline approach that applies direct quantization to the distilled dataset without any additional fine-tuning. This comparison is presented in Table 8. The results clearly demonstrate that BCDC achieves substantially better performance, highlighting the importance of adapting the distilled representations to the quantized setting. By incorporating quantization into the fine-tuning process, BCDC effectively mitigates the performance degradation typically caused by direct quantization, thereby preserving the utility of the condensed data under limited bit precision.

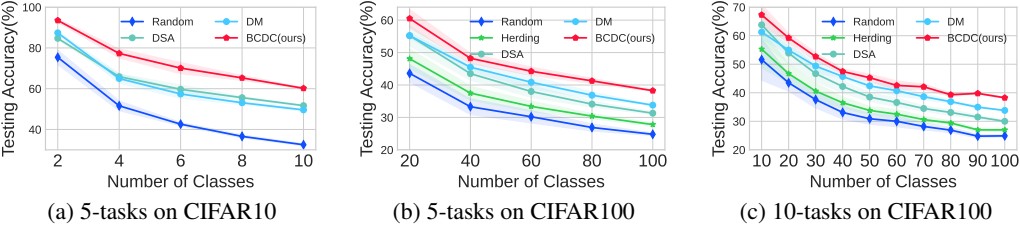

(a) 5-tasks on CIFAR10    (b) 5-tasks on CIFAR100    (c) 10-tasks on CIFAR100

Figure 2: Test accuracy on the class-incremental learning task.

Table 6: Comparison of dataset condensation methods on Tiny-ImageNet and ImageNet-1K. Results are reported as test accuracy (%) on condensed dataset.

| Dataset | Img/Cls | D$^4$M | RDED | CMI | DWA | +BCDC | D$^3$HR | +BCDC |
|---|---|---|---|---|---|---|---|---|
| Tiny-ImageNet | 50 | 46.2 | $58.2 \pm 0.1$ | $53.7 \pm 0.3$ | $52.8 \pm 0.2$ | $55.1 \pm 0.3$ | $56.9 \pm 0.2$ | $59.3 \pm 0.4$ |
| Tiny-ImageNet | 100 | 51.4 | — | $56.9 \pm 0.3$ | $56.0 \pm 0.2$ | $59.6 \pm 0.3$ | $59.3 \pm 0.1$ | $61.8 \pm 0.2$ |
| ImageNet-1K | 10 | 27.9 | $42.0 \pm 0.1$ | $38.5 \pm 0.3$ | $37.9 \pm 0.2$ | $39.6 \pm 0.3$ | $44.3 \pm 0.3$ | $46.9 \pm 0.5$ |
| ImageNet-1K | 50 | 55.2 | $56.5 \pm 0.1$ | $55.6 \pm 0.3$ | $55.2 \pm 0.2$ | $57.5 \pm 0.3$ | $59.4 \pm 0.1$ | $62.6 \pm 0.3$ |
| ImageNet-1K | 100 | 59.3 | — | $59.8 \pm 0.4$ | $59.2 \pm 0.3$ | $61.7 \pm 0.4$ | $62.5 \pm 0.0$ | $64.7 \pm 0.2$ |

Table 7: Cross-architecture evaluation on CIFAR-10 using 10 images per class. *Train* denotes the architecture used to condense the dataset, while *Transfer* refers to the architecture trained on the condensed data.

| Method | Train \ Transfer | ConvNet | LeNet | AlexNet | VGG11 | ResNet18 |
|---|---|---|---|---|---|---|
| DSA | AlexNet | 30.4±0.7 | 24.2±0.4 | 28.3±0.4 | 27.2±1.0 | 27.8±1.1 |
| | ConvNet | 31.4±1.1 | 21.7±1.6 | 25.9±0.8 | 27.6±0.8 | 27.6±1.4 |
| DM | AlexNet | 41.4±0.8 | 31.4±0.2 | 37.5±0.9 | 36.8±0.5 | 34.9±1.1 |
| | ConvNet | 42.2±0.5 | 33.4±0.6 | 38.8±1.3 | 36.2±1.0 | 34.6±0.5 |
| DM+BCDC (ours) | AlexNet | 57.6±0.5 | 33.8±0.7 | 51.2±0.6 | 52.3±0.7 | 52.1±0.8 |
| | ConvNet | 58.0±0.5 | 46.9±0.8 | 54.6±0.9 | 52.0±0.7 | 51.8±0.9 |

**Comparison with standard image compression techniques (JPEG, WebP.)** We compare our approach with standard image compression methods, such as JPEG and WebP, in Table 11 in Appendix. Notably, our method can be seamlessly combined with these compression techniques and yields substantial additional improvements in dataset distillation performance.

**Dataset Condensation Visualization** We present visualizations of the condensed images under different quantization levels integrating with DC in Figure 1 and integrating with DM in Figure 3 in Appendix. As the number of bits used to represent each image increases, more fine-grained visual details are preserved, resulting in higher image quality. However, this comes at the cost of increased storage requirements per image. Conversely, using fewer bits reduces the fidelity of the image, leading to some loss of detail, but significantly lowers the memory footprint. This trade-off enables the storage of a larger number of images within the same memory budget, which is particularly beneficial in resource-constrained scenarios.

**Hyperparameter Sensitivity** We present hyperparameter sensitivity, e.g., $\lambda$, in Appendix D.

## 5.3 DATASET CONDENSATION FOR CONTINUAL LEARNING

In this section, we evaluate the effectiveness of BCDC in the context of continual learning. We adopt a class-incremental learning setting under tight memory constraints—specifically, 10 images per class for CIFAR10 and 20 images per class for CIFAR100. We integrate BCDC with the GDumb framework (Prabhu et al., 2020) using either coreset selection strategies (Random, Herding) or dataset distillation techniques (DSA, DM). Experiments are conducted on two standard benchmarks: CIFAR10, split into 5 sequential tasks, and CIFAR100, evaluated under both 5-task and 10-task settings. As shown in Fig. 2, GDumb combined with BCDC consistently outperforms all other variants. This demonstrates that our low-bit condensed data remains highly informative and effective in CL scenarios.

## 6 CONCLUSION

In this paper, we proposed a low-bit data representation quantization method to compress datasets into small-scale condensed versions, significantly reducing memory storage costs. Through extensive experiments across multiple datasets and settings, we demonstrated the effectiveness of our approach in maintaining data utility while achieving substantial compression. The results highlight the potential of our method to facilitate efficient data storage and processing in resource-constrained environments. Future work could explore adaptive quantization strategies to further enhance performance.

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

**Appendix**

## A DERIVATIONS FOR DATA-GRADIENTS

When adapting parameters $\boldsymbol{\theta}$ to a task via fine-tuning, the optimal post-adaptation parameters $Q(\mathcal{S})$ depend implicitly on $\boldsymbol{\theta}$. We compute $\frac{\partial Q(\mathcal{S})}{\partial \boldsymbol{\theta}}$ using chain rule on the optimality condition:

**Fine-Tuning (Inner Loop)** At convergence, the gradient of the fine-tuning loss $L_{\text{train}}$ w.r.t. $Q(\mathcal{S})$ is zero:

$$\frac{\partial L_{\text{train}}(\boldsymbol{\theta})}{\partial \boldsymbol{\theta}} = 0 \quad \text{(Optimality condition)}$$

Differentiate the optimality condition w.r.t. $\boldsymbol{\theta}$:

$$\frac{\partial}{\partial Q(\mathcal{S})} \left( \frac{\partial L_{\text{train}}(\boldsymbol{\theta})}{\partial \boldsymbol{\theta}} \right) = 0$$

$$\frac{\partial^2 L_{\text{train}}}{\partial Q(\mathcal{S}) \partial \boldsymbol{\theta}} + \frac{\partial^2 L_{\text{train}}}{\partial \boldsymbol{\theta}^2} \frac{\partial \boldsymbol{\theta}}{\partial Q(\mathcal{S})} = 0$$

**Solve for** $\frac{\partial \boldsymbol{\theta}}{\partial Q(\mathcal{S})}$

$$\frac{\partial \boldsymbol{\theta}}{\partial Q(\mathcal{S})} \approx - \left( \frac{\partial^2 L_{\text{train}}}{\partial \boldsymbol{\theta}^2} \right)^{-1} \frac{\partial^2 L_{\text{train}}}{\partial Q(\mathcal{S}) \partial \boldsymbol{\theta}}$$

where $\frac{\partial^2 L_{\text{train}}}{\partial Q(\mathcal{S})^2}$: Hessian of the fine-tuning loss; $\frac{\partial^2 L_{\text{train}}}{\partial Q(\mathcal{S}) \partial \boldsymbol{\theta}}$: Mixed partial derivative; The inverse Hessian adjusts the meta-gradient for inner-loop dynamics.

## B THEOREM PROOF

### B.1 PROOF FOR THEOREM 4.5

*Proof.* We decompose the true risk $R_{\mathcal{D}}(h)$ as:

$$R_{\mathcal{D}}(h) = \underbrace{\hat{R}_{Q_b(\mathcal{S})}(h)}_{\text{Empirical Risk}} + \underbrace{R_{\mathcal{D}}(h) - R_{\mathcal{S}}(h)}_{\text{Condensation Error}} + \underbrace{R_{\mathcal{S}}(h) - \hat{R}_{Q_b(\mathcal{S})}(h)}_{\text{Quantization Error}} \tag{11}$$

Using Rademacher complexity and Hoeffding's inequality:

$$R_{\mathcal{D}}(h) - R_{\mathcal{S}}(h) \le 2\mathfrak{R}_{m(b)}(\mathcal{H}) + \sqrt{\frac{\log(2/\delta)}{2m(b)}} \tag{12}$$

$$\le \frac{2\kappa}{\sqrt{m(b)}} + \sqrt{\frac{\log(2/\delta)}{2m(b)}} \tag{13}$$

**Rademacher Complexity Foundation** For any hypothesis class $\mathcal{H}$, the generalization gap can be bounded via Rademacher complexity:

$$R_{\mathcal{D}}(h) - R_{\mathcal{S}}(h) \le 2\mathfrak{R}_m(\mathcal{H}) + \sup_{h \in \mathcal{H}} |R_{\mathcal{D}}(h) - R_{\mathcal{S}}(h)| \tag{14}$$

Where $\mathfrak{R}_m(\mathcal{H}) = \mathbb{E}_{\mathcal{S}} \mathbb{E}_{\sigma} \left[ \sup_{h \in \mathcal{H}} \frac{1}{m} \sum_{i=1}^{m} \sigma_i \ell(h(x_i^s), y_i^s) \right]$ with $\sigma_i \in \{\pm 1\}$ being Rademacher variables.

**Finite-Sample Concentration**  Applying Hoeffding's inequality to the second term, for any fixed $h$:

$$\mathbb{P}\left(|R_{\mathcal{D}}(h) - R_{\mathcal{S}}(h)| \geq \epsilon\right) \leq 2\exp\left(-\frac{2m\epsilon^2}{L^2}\right) \tag{15}$$

Taking a union bound over $\mathcal{H}$ with finite VC-dimension $d$:

$$\mathbb{P}\left(\sup_{h\in\mathcal{H}} |R_{\mathcal{D}}(h) - R_{\mathcal{S}}(h)| \geq \epsilon\right) \leq 2\mathcal{N}(\mathcal{H}, \epsilon)\exp\left(-\frac{2m\epsilon^2}{L^2}\right) \tag{16}$$

Where $\mathcal{N}(\mathcal{H}, \epsilon)$ is the covering number. For parametric models, $\log\mathcal{N}(\mathcal{H}, \epsilon) \asymp d\log(1/\epsilon)$.

**Solving for High Probability Bound**  Set the RHS to $\delta/2$ and solve for $\epsilon$:

$$2\exp\left(d\log(1/\epsilon) - \frac{2m\epsilon^2}{L^2}\right) = \delta/2 \tag{17}$$

$$\Rightarrow \epsilon \asymp \sqrt{\frac{d\log(m/d) + \log(2/\delta)}{m}} \tag{18}$$

This gives the standard generalization bound:

$$R_{\mathcal{D}}(h) - R_{\mathcal{S}}(h) \leq \frac{2\kappa}{\sqrt{m}} + \sqrt{\frac{\log(2/\delta)}{2m}} \tag{19}$$

Using Lipschitz and $\beta$-smoothness properties:

$$|R_{\mathcal{S}}(h) - \hat{R}_{Q_b(\mathcal{S})}(h)| \leq L\mathbb{E}[\|h(\boldsymbol{x}) - h(Q_b(\boldsymbol{x}))\|] + \frac{\beta}{2}\mathbb{E}[\|h(\boldsymbol{x}) - h(Q_b(\boldsymbol{x}))\|^2] \tag{20}$$

$$\leq LC2^{-b} + \beta C^2 2^{-2b} \tag{21}$$

Applying the union bound and rescaling $\delta$:

$$R_{\mathcal{D}}(h) \leq \hat{R}_{Q_b(\mathcal{S})}(h) + \frac{2\kappa}{\sqrt{m(b)}} + \sqrt{\frac{\log(2/\delta)}{2m(b)}} + LC2^{-b} + \beta C^2 2^{-2b} \tag{22}$$

The following is a detailed derivation of how Rademacher complexity and Hoeffding's inequality are applied in the proof with probabilistic bounds and their interaction with quantization:

**Incorporating Quantization Effects**  When replacing $\mathcal{S}$ with $Q_b(\mathcal{S})$, the key modification appears in the Rademacher term:

$$\mathfrak{R}_m(\mathcal{H}, Q_b) = \mathbb{E}\left[\sup_h \frac{1}{m}\sum_i \sigma_i \ell(h(Q_b(x_i^s)), y_i^s)\right] \tag{23}$$

$$\leq \mathfrak{R}_m(\mathcal{H}) + L\mathbb{E}\left[\sup_h \frac{1}{m}\sum_i \sigma_i(h(Q_b(x_i^s)) - h(x_i^s))\right] \tag{24}$$

$$\leq \mathfrak{R}_m(\mathcal{H}) + LC2^{-b}\sqrt{\frac{d}{m}} \tag{25}$$

The second inequality uses the Lipschitz property and the quantization error bound.

**Final Composition**  Combining all terms while accounting for the memory constraint $m(b) = M/(bd)$:

$$R_{\mathcal{D}}(h) \leq \hat{R}_{Q_b(\mathcal{S})}(h) + \frac{2\kappa}{\sqrt{m(b)}} + \sqrt{\frac{\log(2/\delta)}{2m(b)}} + LC2^{-b} + \beta C^2 2^{-2b} \tag{26}$$

$\square$

## C  DERIVATION OF KL DIVERGENCE BOUND

The Fisher Information Matrix (FIM) is defined as the covariance of the score function (gradient of log-posterior):

$$\mathbf{F} = I(\mathcal{S}; \boldsymbol{\theta}) = \mathbb{E}_{p(\boldsymbol{\theta}|\mathcal{S})} \left[ \nabla_{\boldsymbol{\theta}} \log p(\boldsymbol{\theta}|\mathcal{S}) \nabla_{\boldsymbol{\theta}} \log p(\boldsymbol{\theta}|\mathcal{S})^\top \right] \tag{27}$$

This matrix captures the **local curvature** of the log-posterior, measuring how sensitive the distribution is to small changes in $\boldsymbol{\theta}$.

**Lemma C.1.**

$$D_{KL} \leq \frac{L^2 \Delta^2}{8} \operatorname{tr}(\mathbf{F}) \tag{28}$$

*Proof.* Using the Largest Eigenvalue ($\lambda_{\max}$)

$$\Delta^\top \mathbf{F} \Delta \leq \lambda_{\max}(\mathbf{F}) \|\Delta\|^2 \tag{29}$$

This gives:

$$D_{\text{KL}} \leq \frac{1}{2} \lambda_{\max}(\mathbf{F}) \|\Delta\|^2 \tag{30}$$

Since $\operatorname{tr}(\mathbf{F}) = \sum_i \lambda_i \geq \lambda_{\max}(\mathbf{F})$, we can write:

$$\Delta^\top \mathbf{F} \Delta \leq \operatorname{tr}(\mathbf{F}) \|\Delta\|^2 \tag{31}$$

Thus:

$$D_{\text{KL}} \leq \frac{1}{2} \operatorname{tr}(\mathbf{F}) \|\Delta\|^2 \tag{32}$$

If $\|\nabla_{\boldsymbol{\theta}} \log p(\boldsymbol{\theta}|\mathcal{S})\| \leq L$, then:

The perturbation $\Delta$ may be scaled by $L$ (i.e., $\|\Delta\| \leq \frac{L\Delta}{2}$). This leads to:

$$D_{\text{KL}} \leq \frac{L^2 \Delta^2}{8} \operatorname{tr}(\mathbf{F}) \tag{33}$$

Combining these results:

$$D_{\text{KL}} \leq \frac{L^2 \Delta^2}{8} \operatorname{tr}(I(\mathcal{S}; \boldsymbol{\theta})) \tag{34}$$

where $\operatorname{tr}(I(\mathcal{S}; \boldsymbol{\theta}))$ measures total sensitivity, $L$ is the Lipschitz constant, $\Delta$ is the perturbation magnitude

$\square$

### C.1  THEOREM PROOF FOR THEOREM 4.6

We aim to prove:

$$I(\mathcal{T}; \boldsymbol{\theta}) - I(Q_b(\mathcal{S}); \boldsymbol{\theta}) \leq \mathbb{E}_D \left[ D_{\text{KL}}(p(\boldsymbol{\theta}|\mathcal{T}) \| p(\boldsymbol{\theta}|Q_b(\mathcal{S}))) \right],$$

### EXPRESS MUTUAL INFORMATION AS KL DIVERGENCE

Mutual information can be written as:

$$I(\mathcal{T}; \boldsymbol{\theta}) = \mathbb{E}_D \left[ D_{\text{KL}}(p(\boldsymbol{\theta}|\mathcal{T}) \| p(\boldsymbol{\theta})) \right],$$

$$I(Q_b(\mathcal{S}); \boldsymbol{\theta}) = \mathbb{E}_{Q_b(\mathcal{S})} \left[ D_{\text{KL}}(p(\boldsymbol{\theta}|Q_b(\mathcal{S})) \| p(\boldsymbol{\theta})) \right].$$

Since $Q_b(\mathcal{S})$ is a function of $D$, we rewrite:

$$I(Q_b(\mathcal{S}); \boldsymbol{\theta}) = \mathbb{E}_D \left[ D_{\text{KL}}(p(\boldsymbol{\theta}|Q_b(\mathcal{S})) \| p(\boldsymbol{\theta})) \right].$$

$$I(\mathcal{T}; \boldsymbol{\theta}) - I(Q_b(\mathcal{S}); \boldsymbol{\theta}) = \mathbb{E}_D \left[ D_{\text{KL}}(p(\boldsymbol{\theta}|\mathcal{T}) \| p(\boldsymbol{\theta})) - D_{\text{KL}}(p(\boldsymbol{\theta}|Q_b(\mathcal{S})) \| p(\boldsymbol{\theta})) \right].$$

### EXPAND KL DIVERGENCE

Using $D_{\text{KL}}(p \| q) = \mathbb{E}_p \left[ \log \frac{p}{q} \right]$:

$$D_{\text{KL}}(p(\boldsymbol{\theta}|\mathcal{T}) \| p(\boldsymbol{\theta})) = \mathbb{E}_{\boldsymbol{\theta}|\mathcal{T}} \left[ \log \frac{p(\boldsymbol{\theta}|\mathcal{T})}{p(\boldsymbol{\theta})} \right],$$

$$D_{\text{KL}}(p(\boldsymbol{\theta}|Q_b(\mathcal{S})) \| p(\boldsymbol{\theta})) = \mathbb{E}_{\boldsymbol{\theta}|Q_b(\mathcal{S})} \left[ \log \frac{p(\boldsymbol{\theta}|Q_b(\mathcal{S}))}{p(\boldsymbol{\theta})} \right].$$

### REWRITE THE DIFFERENCE INSIDE EXPECTATION

$$D_{\text{KL}}(p(\boldsymbol{\theta}|\mathcal{T}) \| p(\boldsymbol{\theta})) - D_{\text{KL}}(p(\boldsymbol{\theta}|Q_b(\mathcal{S})) \| p(\boldsymbol{\theta}))$$

$$= \mathbb{E}_{\boldsymbol{\theta}|\mathcal{T}} \left[ \log \frac{p(\boldsymbol{\theta}|\mathcal{T})}{p(\boldsymbol{\theta})} - \log \frac{p(\boldsymbol{\theta}|Q_b(\mathcal{S}))}{p(\boldsymbol{\theta})} \right]$$

$$= \mathbb{E}_{\boldsymbol{\theta}|\mathcal{T}} \left[ \log \frac{p(\boldsymbol{\theta}|\mathcal{T})}{p(\boldsymbol{\theta}|Q_b(\mathcal{S}))} \right]$$

$$= D_{\text{KL}}(p(\boldsymbol{\theta}|\mathcal{T}) \| p(\boldsymbol{\theta}|Q_b(\mathcal{S}))).$$

Take Expectation Over $D$

$$I(\mathcal{T}; \boldsymbol{\theta}) - I(Q_b(\mathcal{S}); \boldsymbol{\theta}) = \mathbb{E}_D \left[ D_{\text{KL}}(p(\boldsymbol{\theta}|\mathcal{T}) \| p(\boldsymbol{\theta}|Q_b(\mathcal{S}))) \right].$$

### INEQUALITY FOR STOCHASTIC $Q_b(\mathcal{S})$

For stochastic $Q_b(\mathcal{S})$ (e.g., variational approximations), we have:

$$I(\mathcal{T}; \boldsymbol{\theta} \mid Q_b(\mathcal{S})) \le \mathbb{E}_D \left[ D_{\text{KL}}(p(\boldsymbol{\theta}|\mathcal{T}) \| p(\boldsymbol{\theta}|Q_b(\mathcal{S}))) \right],$$

where $I(\mathcal{T}; \boldsymbol{\theta} \mid Q_b(\mathcal{S})) = I(\mathcal{T}; \boldsymbol{\theta}) - I(Q_b(\mathcal{S}); \boldsymbol{\theta})$. This holds because equality is achieved when $Q_b(\mathcal{S})$ is deterministic in $D$. For stochastic $Q_b(\mathcal{S})$, the KL divergence overcounts discrepancies, making it an upper bound.

$$I(\mathcal{T}; \boldsymbol{\theta}) - I(Q_b(\mathcal{S}); \boldsymbol{\theta}) \le \mathbb{E}_D \left[ D_{\text{KL}}(p(\boldsymbol{\theta}|\mathcal{T}) \| p(\boldsymbol{\theta}|Q_b(\mathcal{S}))) \right]$$

The second-order Taylor expansion of $\log p(\boldsymbol{\theta} \mid Q_b(\mathcal{S}))$ around $\mathcal{S}$ is:

$$\log p(\boldsymbol{\theta} \mid Q_b(\mathcal{S})) \approx \log p(\boldsymbol{\theta} \mid \mathcal{S}) + \nabla_{\mathcal{S}} \log p(\boldsymbol{\theta} \mid \mathcal{S})^\top (Q_b(\mathcal{S}) - \mathcal{S}) + \frac{1}{2}(Q_b(\mathcal{S}) - \mathcal{S})^\top \nabla_{\mathcal{S}}^2 \log p(\boldsymbol{\theta} \mid \mathcal{S})(Q_b(\mathcal{S}) - \mathcal{S}).$$

(35)

The KL divergence between $p(\boldsymbol{\theta} \mid \mathcal{S})$ and $p(\boldsymbol{\theta} \mid Q_b(\mathcal{S}))$ is:

$$D_{KL}(p(\boldsymbol{\theta} \mid \mathcal{S}) \| p(\boldsymbol{\theta} \mid Q_b(\mathcal{S}))) = \mathbb{E}_{p(\boldsymbol{\theta}|\mathcal{S})} \left[ \log p(\boldsymbol{\theta} \mid \mathcal{S}) - \log p(\boldsymbol{\theta} \mid Q_b(\mathcal{S})) \right]. \quad (36)$$

Substituting the Taylor expansion:

$$D_{KL} \approx \mathbb{E}_{p(\boldsymbol{\theta}|\mathcal{S})} \left[ -\nabla_{\mathcal{S}} \log p(\boldsymbol{\theta} \mid \mathcal{S})^\top (Q_b(\mathcal{S}) - \mathcal{S}) - \frac{1}{2}(Q_b(\mathcal{S}) - \mathcal{S})^\top \nabla_{\mathcal{S}}^2 \log p(\boldsymbol{\theta} \mid \mathcal{S})(Q_b(\mathcal{S}) - \mathcal{S}) \right].$$

(37)

- The score function $\nabla_{\mathcal{S}} \log p(\boldsymbol{\theta} \mid \mathcal{S})$ has zero expectation under $p(\boldsymbol{\theta} \mid \mathcal{S})$:

$$\mathbb{E}_{p(\boldsymbol{\theta}|\mathcal{S})} \left[ \nabla_{\mathcal{S}} \log p(\boldsymbol{\theta} \mid \mathcal{S}) \right] = 0. \tag{38}$$

- The Fisher information matrix is defined as:

$$\mathbf{F} = -\mathbb{E}_{p(\boldsymbol{\theta}|\mathcal{S})} \left[ \nabla_{\mathcal{S}}^2 \log p(\boldsymbol{\theta} \mid \mathcal{S}) \right]. \tag{39}$$

Thus, the KL divergence simplifies to:

$$D_{KL} \approx \frac{1}{2} (Q_b(\mathcal{S}) - \mathcal{S})^\top \mathbf{F} (Q_b(\mathcal{S}) - \mathcal{S}). \tag{40}$$

We define $\quad \epsilon = Q_b(\mathcal{S}) - \mathcal{S}$

The KL divergence becomes:

$$D_{\mathrm{KL}} \approx \frac{1}{2} \mathbb{E} \left[ \epsilon^\top \left( -\nabla_{\mathcal{S}}^2 \log p(\boldsymbol{\theta}|\mathcal{S}) \right) \epsilon \right]$$

**Bounding the Terms**:

$$\|\nabla_{\mathcal{S}}^2 \log p(\boldsymbol{\theta}|\mathcal{S})\|_{\mathrm{op}} \leq L^2 \quad \Rightarrow \quad -\nabla_{\mathcal{S}}^2 \log p(\boldsymbol{\theta}|\mathcal{S}) \preceq L^2 I$$

$$\|\epsilon\|_2 = \|\mathcal{S} - Q_b(\mathcal{S})\|_2 \leq \sqrt{d}\Delta/2$$

$$D_{\mathrm{KL}} \leq \frac{\Delta^2}{8} \mathrm{tr} \left( \mathbb{E} \left[ \nabla_{\boldsymbol{\theta}} \log p(\boldsymbol{\theta}|\mathcal{S}) \nabla_{\boldsymbol{\theta}} \log p(\boldsymbol{\theta}|\mathcal{S})^\top \right] \right) = \frac{L^2 \Delta^2}{8} \mathrm{tr}(I(\mathcal{S}; \boldsymbol{\theta}))$$

The Fisher information loss due to $b$-bit quantization is bounded by:

$$I(\mathcal{T}; \boldsymbol{\theta}) - I(Q_b(\mathcal{S}); \boldsymbol{\theta}) \leq \frac{L^2 \Delta^2}{8} \mathrm{tr} \left( \mathbb{E} \left[ \nabla_{\boldsymbol{\theta}} \log p(\boldsymbol{\theta}|\mathcal{S}) \nabla_{\boldsymbol{\theta}} \log p(\boldsymbol{\theta}|\mathcal{S})^\top \right] \right)$$

where $\Delta = 2^{-b+1}(\max(\mathcal{S}) - \min(\mathcal{S}))$.

# D MORE EXPERIMENTAL RESULTS

Table 9: Training Efficiency Comparison of Dataset Distillation Methods

| Method | DC | DC+BCDC | DM | DM+BCDC |
|---|---|---|---|---|
| **Running Time** (hours) | 10.97 | 12.83 | 7.15 | 8.62 |

Table 10: Effect of Bandwidth on Dataset Distillation Performance on CIFAR10 Under the Same Memory Budget

| Bandwidth (bits/sample) | 2-bit | 4-bit | 8-bit |
|---|---|---|---|
| **DC+BCDC (SM)** | 25.2±0.9 | 36.1±0.6 | 27.3±0.7 |
| **DSA+BCDC (SM)** | 28.3±0.6 | 42.2±0.3 | 29.4±0.5 |
| **DM+BCDC (SM)** | 31.7±0.3 | 45.1±0.8 | 30.5±0.6 |

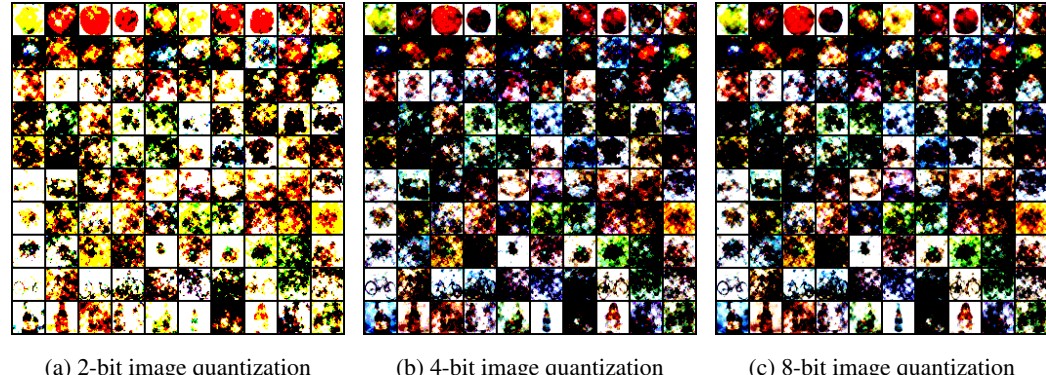

| (a) 2-bit image quantization | (b) 4-bit image quantization | (c) 8-bit image quantization |

Figure 3: Visualization of different bits image quantization by integrating BCDC with DM.

## D.1 COMPARISON WITH STANDARD IMAGE COMPRESSION TECHNIQUES (JPEG, WEBP.)

We compare our approach with standard image compression methods, such as JPEG and WebP, in Table 11. Notably, our method can be seamlessly combined with these compression techniques and yields substantial additional improvements in dataset distillation performance.

| Dataset | Img/Cls | DM | DM+BCDC (Ours) | BCDC+JPEG | BCDC+WebP |
|---|---|---|---|---|---|
| CIFAR100 | 10 | $29.7 \pm 0.3$ | $38.3 \pm 0.7$ | $41.2 \pm 0.6$ | $43.6 \pm 0.8$ |

Table 11: Comparison with standard image compression techniques (JPEG, WebP, etc.) on CIFAR100.

**Dataset Condensation with Distribution Matching By Quantized Data Representations** Integrating BCDC with surrogate loss function, e.g., DM (Zhao & Bilen, 2023) is presented in Algorithm 2.

---

**Algorithm 2** Dataset Condensation with Distribution Matching By Quantized Data Representations

---

1: **Require:** Training set $\mathcal{T}$, randomly initialized synthetic samples $\mathcal{S}$ for $J$ classes, deep neural network $\psi_{\boldsymbol{\theta}}$ parameterized by $\boldsymbol{\theta}$, parameter distribution $P_{\boldsymbol{\theta}}$, differentiable augmentation $\mathcal{A}_{\omega}$ parameterized by $\omega$, augmentation distribution $\Omega$, training iterations $K$, learning rate $\eta$
2: **for** $k = 0$ to $K - 1$ **do**
3:    Sample $\boldsymbol{\theta} \sim P_{\boldsymbol{\theta}}$
4:    **for** each class $j = 0$ to $J - 1$ **do**
5:       Sample mini-batch pairs $B_c^{\mathcal{T}} \sim \mathcal{T}$, $B_c^{\mathcal{S}} \sim \mathcal{S}$, and $\omega_c \sim \Omega$
6:    **end for**
7:    Compute the loss:
   $$\mathcal{L} = \sum_{c=0}^{C-1} \left\| \frac{1}{|B_c^{\mathcal{T}}|} \sum_{(\boldsymbol{x},y) \in B_c^{\mathcal{T}}} \psi_{\boldsymbol{\theta}}(\mathcal{A}_{\omega_c}(\boldsymbol{x})) - \frac{1}{|B_c^{\mathcal{S}}|} \sum_{(\mathbf{s},y) \in B_c^{\mathcal{S}}} \psi_{\boldsymbol{\theta}}(\mathcal{A}_{\omega_c}(Q_V(\mathbf{s}))) \right\|^2$$
8:    Update $\mathcal{S} \leftarrow \mathcal{S} - \eta \nabla_{\mathcal{S}} \mathcal{L}$
9: **end for**
10: **Output:** $\mathcal{S}$

---

