# OpenReview forum: "Toward Bit-Efficient Dataset Condensation: A General Framework"
_ICLR.cc/2026/Conference — Submitted to ICLR 2026_

### Official Review · Reviewer_2oeQ · 2025-10-26

**Soundness:** 3
**Presentation:** 3
**Contribution:** 2
**Rating:** 4
**Confidence:** 4

**Summary:**

This paper looks at "dataset condensation," which is the task of shrinking a giant dataset (like ImageNet) into a tiny set of synthetic images that can be used to train a model just as well. The authors' key insight is that everyone has been focused on shrinking the number of images (e.g., 10 images per class) but has ignored the fact that these images are still stored in full 32-bit precision, which makes them huge in terms of file size. This creates real-world bottlenecks for sending data from the cloud to an edge device or in federated learning.
The paper proposes a general framework to learn a small set of quantized (i.e., low bit-rate) synthetic samples from the start.

**Strengths:**

1.	I find this paper to be original and well-motivated. The authors identified a super practical, high-impact problem that the rest of the field seems to have overlooked. The "bits-per-image" dimension is just as important as the "images-per-class" dimension for any real-world deployment, and this paper is one of the first to tackle it head-on.

2.	The significance here is obvious. A method that produces a dataset that is both small in number and small in file size is exponentially more useful than the current state-of-the-art. The proposed method can be easily combined with other state-of-the-art dataset condensation methods.

3.	The theoretical analysis shows the upper bound of errors under fixed bits.

**Weaknesses:**

1.	Even though the idea of focusing on the total memory size of the condensed dataset is interesting, the technical contribution is relatively low. The quantization-aware training is not new but well-explored.

2.	Some experiments are missing in Table 3 compared to Table 2.

3.	The Comparison with standard image compression techniques is not clear. Could the authors provide more details such as how and where JPEG is applied to?

4.	When the total memory is fixed, it will be interesting to see how the changes of # of images and bits per pixel affect the final accuracy. Could the authors provide some results regarding this?

**Questions:**

See the weaknesses.

---

> ### Author Response · Authors · 2025-11-21
> **Authors Rebuttal**
>
> **W1**  Even though the idea of focusing on the total memory size of the condensed dataset is interesting, the technical contribution is relatively low. The quantization-aware training is not new but well-explored.
>
>
>
> **A1**  Thank you for the thoughtful comment. While quantization-aware training is indeed well studied for model compression, our work tackles a very different problem: quantizing synthetic distilled data. This is not addressed in prior literature, and directly applying existing quantizations to distilled images leads to highly unstable optimization. Our framework, BCDC, introduces mechanisms specifically designed for this new setting, including a quantization-fidelity loss, a differentiable quantizer strategies that make low-bit dataset condensation viable.
>
> BCDC also reframes dataset condensation through a memory-constrained lens. Prior dataset condensation methods focus on reducing the number of synthetic samples but ignore per-sample bit width, representational diversity under fixed memory, and how fidelity should be traded off against sample count. Our formulation exposes this previously overlooked fidelity–diversity frontier and provides both theoretical and empirical insights into how bit-budgeted synthetic data should be optimized.
>
> In short, the novelty lies not in reusing standard QAT, but in addressing a new problem, i.e., quantizing distilled data and developing the technical components needed to make that problem solvable. The resulting framework advances dataset condensation in a direction that has not been explored before.
>
> ---
>
>
> **W2** Some experiments are missing in Table 3 compared to Table 2.
>
> **A2** Thank you for pointing out this. We have now completed Table 3 and ensured that corresponding experimental results are included for consistency with Table 2.
>
> ---
>
> **W3** The Comparison with standard image compression techniques is not clear. Could the authors provide more details such as how and where JPEG is applied to?
>
> **A3**  In this table, we first use our BCDC method to generate a quantized and distilled dataset. Next, we apply JPEG compression to the distilled dataset. The quantized and distilled dataset has the same memory consumption as the full-precision distilled dataset. Finally, we train the model using the resulting compressed dataset.
>
> ---
>
> **W4** When the total memory is fixed, it will be interesting to see how the changes of # of images and bits per pixel affect the final accuracy. Could the authors provide some results regarding this?
>
>
> **A4** Thank you for your constructive suggestions. We provided these results in Table 10 in Appendix D, which analyzes how quantization bit-width affects dataset distillation performance under a fixed memory budget. Lower bit-widths substantially reduce per-image storage cost, allowing more synthetic samples to be retained and thereby improving feature space coverage. However, aggressive quantization, e.g., 2-bit, introduces excessive quantization noise that distorts the synthetic images, leading to weaker supervision signals. As bit-width increases, this distortion decreases and performance improves, reaching an optimal balance around 4-bit quantization. Beyond this point, although the visual fidelity of individual synthetic samples continues to improve, the effective dataset size must shrink to maintain the same memory budget, reducing sample diversity and harming feature coverage. This explains why performance declines at higher bit-widths despite improved per-sample quality. Overall, these observations reveal a fundamental trade-off between per-sample fidelity and dataset diversity, demonstrating that 4-bit quantization provides the best balance for maintaining both memory efficiency and model performance.
>
>
>
> *Table: Effect of bits-per pixels/(number of images) on Dataset Distillation Performance on CIFAR-10 Under the Same Memory Budget (SM)*
>
> | **Bandwidth (bits/sample)** | **2-bit** (16 images/class) | **4-bit** (8 images/class)  | **8-bit** (4 images/class)  |
> | --------------------------- | ---------- | ---------- | ---------- |
> | **DC+BCDC (SM)**            | 25.2 ± 0.9 | 36.1 ± 0.6 | 27.3 ± 0.7 |
> | **DSA+BCDC (SM)**           | 28.3 ± 0.6 | 42.2 ± 0.3 | 29.4 ± 0.5 |
> | **DM+BCDC (SM)**            | 31.7 ± 0.3 | 45.1 ± 0.8 | 30.5 ± 0.6 |
>
> ---

---

### Official Review · Reviewer_gViH · 2025-10-27

**Soundness:** 3
**Presentation:** 2
**Contribution:** 3
**Rating:** 6
**Confidence:** 2

**Summary:**

The paper presents a Bit-Conscious Dataset Condensation (BCDC) framework that integrates a Differentiable Data Quantization (DDQ) module to enable low-bit dataset condensation. The method allows joint optimization of data content and quantized representation, and provides theoretical analysis on the trade-off between quantization and generalization errors under a fixed memory budget. Experimental results on ImageNet-1K, cross-architecture, and continual learning settings demonstrate consistent performance gains within the same memory constraints.

**Strengths:**

1. This paper is the first to explore bit-efficient dataset condensation, addressing a key bottleneck of DC methods in low-resource environments.
2. The introduction of the DDQ module effectively overcomes the non-differentiability of hard quantization through smooth approximation, enabling joint optimization of condensed content and bit representation.
3. The paper provides a rigorous theoretical analysis that establishes a clear trade-off between quantization error and generalization error under memory constraints.

**Weaknesses:**

1. The motivation requires clarification. The paper should more clearly explain the three challenges: bandwidth cost, memory consumption, and hardware efficiency, since dataset condensation itself already reduces these costs. It remains unclear why further quantization is necessary.
2. The discussion of related work is insufficient. The authors should include prior studies connecting quantization and dataset condensation, as this directly motivates the proposed approach of enhancing DC via quantization.
3. The paper mentions that the proposed method fine-tunes full-precision condensed datasets. However, during BCDC optimization, the full-precision synthetic data $\mathcal{S}$ still needs to be stored in memory. Considering that memory efficiency is a key motivation, this design limits memory savings during training, with benefits only evident in the final quantized dataset $\mathcal{S}_{quant}$. The paper should clarify the actual memory impact during optimization.
4. Experimental results show that performance peaks at 4-bit precision and declines thereafter, mainly due to the reduced number of stored samples. Although the theoretical section discusses the trade-off, the main paper lacks in-depth analysis of the performance degradation beyond the optimal bit width, which should be further explored.
5. The DDQ module introduces parameters $k$ and $s$. The paper lacks a sensitivity analysis and practical guidance for choosing the key smoothing parameter $k$. An inappropriate value of $k$ may lead to inaccurate approximation or unstable training.

**Questions:**

See weakness.

---

> ### Author Response · Authors · 2025-11-21
> **Authors Rebuttal (1/3)**
>
> **W1** The motivation requires clarification. The paper should more clearly explain the three challenges: bandwidth cost, memory consumption, and hardware efficiency, since dataset condensation itself already reduces these costs. It remains unclear why further quantization is necessary.
>
> **A1** Thank you for raising this important point. We clarify that dataset condensation and quantization solve different bottlenecks. While condensation reduces the number of samples, it preserves their precision. As a result, many practical constraints—bandwidth limitations, memory pressure, and suboptimal hardware utilization—remain unresolved even when the dataset size becomes small. Bit-efficient condensation is therefore not redundant, but necessary to address per-sample efficiency that condensation alone cannot provide.
>
> First, condensation reduces sample count but not cost per sample: FP32 synthetic images still incur high communication cost, I/O latency, and memory overhead. In distributed, edge, or federated settings, transmitting or storing FP32 condensed samples remains a dominant bottleneck. Moreover, FP32 inputs prevent accelerators such as Tensor Cores, TPUs, and mobile AI engines from reaching peak throughput, because these architectures are optimized for lower-precision data. Thus, even a small FP32 distilled dataset can significantly underutilize modern hardware.
>
> Second, quantization directly addresses these remaining challenges by lowering the precision of each stored sample. Low-bit condensed data substantially reduces transmission size, GPU/CPU memory usage, cache pressure, and data-loading latency—complementing condensation's reduction in sample count. It also allows the condensed dataset to align with low-precision execution pipelines (INT8/INT4 kernels), enabling more efficient downstream training than is possible with FP32 inputs.
>
> In summary, condensation and quantization form **two complementary axes of efficiency: condensation compresses how many samples are stored, while quantization compresses how much information each sample carries**. This joint compression is essential for practical deployment, especially under bandwidth constraints, memory limitations, or low-precision hardware. Our experiments also demonstrate that low-bit condensed data not only reduces resource consumption but also improves downstream performance under fixed budgets—showing the necessity of integrating quantization into dataset condensation.
>
> ---
>
> **W2**  The discussion of related work is insufficient. The authors should include prior studies connecting quantization and dataset condensation, as this directly motivates the proposed approach of enhancing DC via quantization.
>
>
> **A2**: Thank you for this valuable suggestion. We emphasize that no existing work integrates quantization into dataset condensation (DC). All prior DC methods store synthetic samples in full precision and optimize them without considering bit budgets, bandwidth constraints, or quantization-aware objectives. Conversely, work on quantization focuses exclusively on model weights or activations—post-training quantization, or low-bit training—but does not address quantizing synthetic datasets or incorporating quantization constraints.
>
> This lack of connection between the two areas is precisely the motivation for our BCDC framework. Condensation reduces the number of samples but does not reduce per-sample precision; quantization methods improve model efficiency but do not improve synthetic dataset efficiency. No prior method jointly optimizes synthetic data and their quantized representations or provides a differentiable mechanism to handle discrete bit-width constraints.
>
> In summary, our BCDC is the first to bridge these two previously disjoint research directions by introducing quantization-aware optimization into dataset condensation. We have updated the related work section in the revision to explicitly highlight this gap and clarify how BCDC fills it.

---

> ### Author Response · Authors · 2025-11-21
> **Authors Rebuttal (2/3)**
>
> **W3**  The paper mentions that the proposed method fine-tunes full-precision condensed datasets. However, during BCDC optimization, the full-precision synthetic data
>  still needs to be stored in memory. Considering that memory efficiency is a key motivation, this design limits memory savings during training, with benefits only evident in the final quantized dataset
>  . The paper should clarify the actual memory impact during optimization.
>
>
>
>  **A3**: Thank you for raising this point. We clarify that full-precision synthetic data are required only during the condensation stage. Once condensation is complete, the full-precision synthetic data are immediately discarded. All downstream training uses only the quantized condensed dataset, which provides 4–8× memory savings. Thus, the memory benefits apply exactly where quantized data is repeatedly deployed—on edge devices, federated clients, and resource-limited training environments.
>
>
>
> In practice, the dominant memory savings of dataset condensation arise after condensation. Then, the distilled dataset is stored, transmitted, and repeatedly used for downstream training. Full-precision dataset condensation methods still require storing FP32 images, while BCDC stores low-bit synthetic data, reducing memory and bandwidth across every future training cycle. Hence, while full-precision is used only transiently during optimization, as in all prior works, the substantial memory efficiency of BCDC is realized in the actual deployment phase for which condensed datasets are designed.
>
> ---
>
> **W4** Experimental results show that performance peaks at 4-bit precision and declines thereafter, mainly due to the reduced number of stored samples. Although the theoretical section discusses the trade-off, the main paper lacks in-depth analysis of the performance degradation beyond the optimal bit width, which should be further explored.
>
>
> **A4**  Thank you for your thoughtful and helpful suggestions. We agree that understanding why performance peaks at a certain bit-precision is essential for characterizing the practical limits of BCDC. Below we clarify the underlying reasons and expand the analysis.
>
> Table 10 in Appendix D in our original submission (we also provide the table in the rebuttal in the following) analyzes how quantization bit-width affects dataset distillation performance under a fixed memory budget. Lower bit-widths substantially reduce per-image storage cost, allowing more synthetic samples to be retained and thereby improving feature-space coverage during model training. However, aggressive quantization introduces excessive quantization noise that distorts the synthetic images, leading to weaker supervision signals. As bit-width increases, this distortion decreases and performance improves, reaching an optimal balance around 4-bit quantization. Beyond this point, although the visual fidelity of individual synthetic samples continues to improve, the effective dataset size must shrink to maintain the same memory budget, reducing sample diversity and harming feature coverage—two factors empirically more critical than marginal fidelity gains. This explains why performance declines at higher bit-widths despite improved per-sample quality. Overall, these observations reveal a fundamental trade-off between per-sample fidelity and dataset diversity, demonstrating that 4-bit quantization provides the best balance for maintaining both memory efficiency and model performance.
>
>
> | **Bandwidth (bits/sample)** | **2-bit**      | **4-bit**      | **8-bit**      |
> |-----------------------------|----------------|----------------|----------------|
> | **DC+BCDC (SM)**            | 25.2±0.9       | 36.1±0.6       | 27.3±0.7       |
> | **DSA+BCDC (SM)**           | 28.3±0.6       | 42.2±0.3       | 29.4±0.5       |
> | **DM+BCDC (SM)**            | 31.7±0.3       | 45.1±0.8       | 30.5±0.6       |

---

> ### Author Response · Authors · 2025-11-22
> **Authors Rebuttal (3/3)**
>
> **W5** The DDQ module introduces parameters $k$ and $s$. The paper lacks a sensitivity analysis and practical guidance for choosing the key smoothing parameter $k$. An inappropriate value of $k$ may lead to inaccurate approximation or unstable training.
>
>
> **A5** Thank you for your constructive feedback.  We would like to clarify that $s$ is determined by $k$, so only $k$ need to be selected. The sharpness parameter $k$ in the differentiable quantization module determines how closely the soft quantizer approximates a discrete low-bit quantization function. A larger $k$ yields a sharper, more stepwise transition, while a smaller $k$ corresponds to a smoother relaxation.
>
>
> Rather than tuning $k$ directly, we tune $c = k\Delta$, where $\Delta$ is the width between quantization bin centers. This scaling allows the sharpness to be interpreted independently of bin size and provides a more consistent behavior across bit widths. Empirically, we find that quantizers with $c\in[2,6]$ achieve an effective balance between gradient flow and quantization accuracy. The effect of $c$ is shown in the following table:
>
>  **Summary of Quantizer Sharpness via ($c = k\Delta$), where $k= \frac{c}{\Delta}$**
>
> | $c$ | Edge Value ($u = \frac{c}{2}$) | ($\tanh(u)$) at Bin Edges  | Saturation Level           | Interpretation                                                           |
> | ------------- | -------------------- | ------------------------ | -------------------------- | ------------------------------------------------------------------------ |
> | 2         | (u = 1)              | ($\tanh(1) \approx 0.76$)  | Soft                   | Bins are blurred; wide smooth transition.              |
> | 4         | (u = 2)              | ($\tanh(2) \approx 0.964$) | ~96% saturated         | Good trade-off: step-like but still trainable. |
> | 6         | (u = 3)              | ($\tanh(3) \approx 0.995$) | Nearly fully saturated | Almost hard quantization.           |
>
>
> * **($c \approx 2$)** indicates too soft; quantization overly smooth.
> * **($c \approx 4$)** indicates real quantization but maintains useful gradients.
> * **($c \approx 6$)** indicates too hard; tanh saturates.
>
>
>
> We adopt a cosine annealing schedule to control the sharpness parameter $c$ during condensation. The sharpness evolves smoothly from a soft regime $c_{\min}$ to an almost-hard regime $c_{\max}$ according to
>
> $c(t) = c_{\min} + (c_{\max} - c_{\min})\left(1 - \cos\left(\frac{\pi t}{2T}\right)\right), \qquad 0 \le t \le T$
>
> which naturally indicates stable gradients at the beginning of training and gradually aligns the soft relaxation with the target low-bit quantization.
>
> This design is motivated by the observation that early-stage condensation benefits from a smooth quantizer with well-behaved gradients, while later stages require a sharper quantizer to approximate discrete quantization. The cosine schedule increases $c$ slowly at first, accelerates in the middle, and slows again near convergence—avoiding abrupt transitions and preventing gradient saturation. Overall, the cosine schedule offers a simple and effective mechanism for soft-to-hard quantization, ensuring stable optimization while progressively enforcing low-bit constraints. We restrict the adjustment of $c$ to be within ($c\in[2,6]$). These values loosely correspond to *soft*, *balanced*, and *almost-hard* quantization regimes. Moving within this small grid allows the quantization sharpness to adapt to changing training dynamics.

---

### Official Review · Reviewer_M8e6 · 2025-10-30

**Soundness:** 3
**Presentation:** 2
**Contribution:** 2
**Rating:** 4
**Confidence:** 4

**Summary:**

The paper introduces a novel and interesting problem to the dataset condensation community: optimizing condensed datasets for bit-efficiency. The authors propose the Bit-Conscious Dataset Condensation (BCDC) framework, which integrates a differentiable data quantization (DDQ) mechanism into the optimization loop. The core hypothesis is that under a fixed memory budget, storing a larger number of low-fidelity (low-bit) synthetic samples is more effective for training a downstream model than storing a smaller number of full-precision samples. The paper provides theoretical analysis and extensive experiments to support this claim, showing significant performance improvements over state-of-the-art methods when memory constraints are enforced.

**Strengths:**

- The paper's primary strength is its originality. It shifts the paradigm of dataset condensation from merely minimizing the number of samples to optimizing the total information density within a given bit budget.
-  The experiments convincingly demonstrate the effectiveness of BCDC under the "Same Memory" (SM) condition. The performance gains shown in Tables 3, 4, 5, and 6 are substantial and provide strong evidence for the paper's central hypothesis on the tested datasets. The ablation study in Table 8 successfully highlights the importance of the joint optimization approach over a naive post-hoc quantization baseline.

**Weaknesses:**

- The paper claims a "modest increase in training cost—ranging from 17% to 20%" (Sec 5.2, Table 9). This analysis is insufficient and potentially misleading. The complexity of the DDQ process, particularly the backpropagation through the  `tanh`  approximation and the additional quantization loss term, is not constant. The paper provides no analysis of how this overhead scales with data dimensionality (e.g., higher-resolution images) or the number of quantization levels (i.e., bit-depth). A 20% overhead on CIFAR-10 could easily become a prohibitive bottleneck on more complex, real-world data, which challenges the method's claimed efficiency.
- Is this trade-off of per-sample fidelity favorable? For tasks that depend heavily on high-frequency information, such as fine-grained classification (e.g., distinguishing bird species) or medical image analysis where subtle textural anomalies are crucial, this specific form of information loss could be disastrous. The paper's claims of general improvement are unsubstantiated because the "richer information" from more samples may not be the right information for all tasks.
- The BCDC framework introduces new hyperparameters, most notably the quantization loss weight \lambda  and the  tanh  sharpness parameter  k . The main paper fixes \lambda = 0.2  and relegates a brief sensitivity discussion to the appendix, which is insufficient for a core component of the optimization objective. The method's performance likely hinges on a careful balancing of the validation and quantization losses, governed by $\lambda$ .

**Questions:**

Please refer weakness section.

---

> ### Author Response · Authors · 2025-11-21
> **Authors Rebuttal (1/2)**
>
> **W1** The paper claims a "modest increase in training cost—ranging from 17% to 20%" (Sec 5.2, Table 9). This analysis is insufficient and potentially misleading. The complexity of the DDQ process, particularly the backpropagation through the  tanh  approximation and the additional quantization loss term, is not constant. The paper provides no analysis of how this overhead scales with data dimensionality (e.g., higher-resolution images) or the number of quantization levels (i.e., bit-depth). A 20% overhead on CIFAR-10 could easily become a prohibitive bottleneck on more complex, real-world data, which challenges the method's claimed efficiency.
>
> **A1** Thank you for pointing out this! We performed additional experiments on ImageNet with image resolution of 224×224 with ResNet-18 as backbone network and conduct experiment on SRe$^2$L [1] when generating one image with one iteration update on synthetic data on A6000 GPU as shown in the following table. The synthesis time only increases less than 16%
>
> *Synthesis Time Efficiency Comparison of Dataset Distillation Methods on ImageNet with ResNet18 as backbone network*
>
> | **Method** | **SRe$^2$L** | **SRe$^2$L+BCDC** |
> |------------|-------:|------------:|
> | **Running Time (ms)** | 21.82 | 25.19 |
>
>
>
> Reference:
>
> [1] Squeeze, Recover and Relabel: Dataset Condensation at
> ImageNet Scale From A New Perspective​, NeurIPS 2023
>
>
> **W2** Is this trade-off of per-sample fidelity favorable? For tasks that depend heavily on high-frequency information, such as fine-grained classification (e.g., distinguishing bird species) or medical image analysis where subtle textural anomalies are crucial, this specific form of information loss could be disastrous. The paper's claims of general improvement are unsubstantiated because the "richer information" from more samples may not be the right information for all tasks.
>
>
> **A2**  Thank you for raising this concern. While it is true that quantization reduces per-sample fidelity, our experiments consistently show that increasing the number of synthetic samples compensates for this loss by better capturing intra-class diversity. Moreover, the bit width is fully adjustable: for tasks that rely on fine textures or high-frequency cues, users can simply choose higher precision (e.g., 8-bit), so the method does not force low-fidelity setting.
>
> It is also important to clarify that **dataset condensation is not a reconstruction problem**. **Synthetic images are not meant to retain the pixel-level detail or textures of the original data; they aim to encode task-relevant structure**. Prior work [1] shows that condensed samples are often abstract or unrealistic, which is desirable to protect data privacy. In fact, lower-bit representations can even reinforce desirable properties such as privacy, since they avoid reproducing identifiable fine-grained features from the training data.
>
> Overall, we would like to emphasize that the fidelity preservation is not the goal of condensation. The objective is to capture the information needed for learning, not to maintain high-frequency visual detail. Quantized condensation remains effective and flexible, and higher-precision modes are available when a task genuinely requires them.
>
> Reference:
>
> [1] Dataset Distillation via Factorization, NeurIPS 2022

---

> > ### Author Response · Authors · 2025-11-21
> > **Authors Rebuttal (2/2)**
> >
> > **W3** The BCDC framework introduces new hyperparameters, most notably the quantization loss weight $\lambda$  and the  tanh  sharpness parameter  k . The main paper fixes $\lambda$ = 0.2  and relegates a brief sensitivity discussion to the appendix, which is insufficient for a core component of the optimization objective. The method's performance likely hinges on a careful balancing of the validation and quantization losses, governed by
> >  $\lambda$.
> >
> >  **A3**: We appreciate your comments! We provide sensitivity analysis for $\lambda$:
> >
> >
> >  *Table: Sensitivity of $\lambda$ on Condensation Performance with DM+BCDC on CIFAR100*
> >
> >
> >
> > |  | Img/Cls | $\lambda=0.0$ |$\lambda=0.1$  |$\lambda=0.2$  |$\lambda=0.5$ |
> > |------|------|------|------|------|------|
> > |CIFAR100  | 1 | 22.5 $\pm$ 0.7  | 24.9 $\pm$ 0.6 | 26.3 $\pm$ 0.5 | 25.2 $\pm$ 0.4 |
> > |  | 10 | 33.7 $\pm$ 0.8 | 36.9 $\pm$ 0.9 | 38.3 $\pm$ 0.7 | 39.5 $\pm$ 0.6 |
> >
> > The hyperparameter $\lambda$ plays a central role in balancing the validation loss and the quantization loss during dataset distillation. When $\lambda$ is too small, the optimization is dominated by the validation loss, causing the synthetic samples to focus solely on maximizing task performance in a full-precision environment. Although such samples may encode rich class-specific information, they are not robust to the low-bit quantization applied during training or deployment. As a result, their representational fidelity deteriorates substantially after quantization, leading to degraded downstream accuracy. In contrast, when $\lambda$ is excessively large, the synthetic samples are forced to adhere too strictly to quantization constraints. This over-regularization restricts their expressiveness and diversity, preventing them from capturing essential discriminative structures. The resulting samples become overly rigid and lose their capacity to effectively learn useful signals.
> >
> > Between these extremes lies a moderate $\lambda$ region where dataset distillation achieves its best performance. In this regime, the validation loss remains sufficiently influential to ensure that synthetic samples encode meaningful and discriminative patterns, while the quantization loss provides just enough guidance to make these samples robust to the low-precision representation. This balanced trade-off allows the distilled dataset to retain high informational content while reducing storage requirements.

---

### Official Review · Reviewer_uiJF · 2025-10-31

**Soundness:** 3
**Presentation:** 3
**Contribution:** 2
**Rating:** 4
**Confidence:** 4

**Summary:**

This paper explores the problems of dataset distillation in resource-constrained conditions. Distilled datasets typically adopt full-precision representations. However, under resource-constrained condition, it will lead to transmission bottlenecks, excessive memory overhead, and insufficient hardware utilization issues. This paper proposes BCDC framework, applying low-bit quantization to achieve efficient data storage and training in resource-constrained conditions. It can be applied to the existing dataset distillation method. Specifically, BCDC introduces a differentiable bit-aware optimization algorithm to fine-tune the distilled full-precision data into a low-bit representation, allowing for the storage of more synthetic data within the same memory budget. This paper also theoretically analyzes the trade-off between quantization error and generalization error, and also the preservation of Fisher information. BCDC can be applied to existing dataset distillation method with a slight performance drop.

**Strengths:**

1. This paper presents a clear problem and motivation. It tackles the limitations of current dataset distillation in resource-constrained settings by introducing low-bit quantization to improve both storage and computational efficiency.
2. This paper provides a detailed theoretical analysis of the trade-off between quantization error and generalization error under fixed memory constraints, and gives the limits for Fisher information preservation under bit compression.
3. This paper is well-written with clear presentation.

**Weaknesses:**

1. As the proposed method adopt low-bit quantization to reduce the storage requirements for distilled images, more samples can be stored with the same memory budget. It is known that more training distilled samples will improve downstream performance under dataset distillation settings. However, more training samples also increase downstream training time, which actually contradicts to training efficiency purpose of dataset distillation.
2. This paper does not explicitly state whether the quantized synthetic data is processed in low-bit form during downstream training. From the description, differentiability appears to be primarily used as a soft approximation in the optimization process, but how it is processed during downstream training is not stated in detail.
3. This paper claims that distilled datasets in low-bit formats can improve hardware utilization, but no empirical evidence or explanation is provided to support the claims.
4. This paper only compares the performance with full-precision dataset distillation methods, neglecting distilled data parameterization methods. Distilled data parameterization methods store the distilled data in a low-dimensional or implicit form, which can also reduce storage costs. This weakens the fairness of the evaluation and makes it difficult to demonstrate the effectiveness of the proposed method.

**Questions:**

1. Given the same downstream training time budget, how do storage and downstream performance compared with the full-precision method?
2. It would be helpful if the authors could provide implementation details for downstream training and how the proposed method improves hardware utilization.
3. The authors could consider briefly discussing how their proposed method differs from data parameterization dataset distillation methods based on latent variables, and may include them in a baseline comparison.
4. Can this method be used for the distilled data generated from other tasks?

---

> ### Author Response · Authors · 2025-11-21
> **Authors Rebuttal (1/3)**
>
> **W1** As the proposed method adopt low-bit quantization to reduce the storage requirements for distilled images, more samples can be stored with the same memory budget. It is known that more training distilled samples will improve downstream performance under dataset distillation settings. However, more training samples also increase downstream training time, which actually contradicts to training efficiency purpose of dataset distillation.
>
>
> **A1** Thank you for your thoughtful comments. While it's true that increasing the number of distilled samples may lengthen downstream training, the use of low-bit quantized synthetic data offers a different efficiency trade-off. The primary goal of dataset distillation is not only to reduce training time but also to minimize memory/storage overhead without sacrificing model performance. By adopting low-bit quantization, our method enables **storing more informative samples within the same memory budget**, which can **boost accuracy without requiring more high-precision memory**.
>
> Moreover, the increase in downstream training time due to more samples is **marginal** compared to training on full datasets (since the number of distilled data points is much less than the size of the original entire dataset) and is often outweighed by the **significant gains in storage efficiency** and **deployment performance improvement**—particularly in resource-constrained settings (e.g., edge devices, mobile platforms). Importantly, quantized samples are computationally cheaper to process per iteration due to their compact representation, potentially offsetting the cost of having more samples.
>
> In short, our method expands the efficiency frontier by offering a new trade-off: **slightly increased training time for signficantly improved accuracy**. This aligns well with practical deployment scenarios where model performance is critical.
>
> ---
>
>
> **W2** This paper does not explicitly state whether the quantized synthetic data is processed in low-bit form during downstream training. From the description, differentiability appears to be primarily used as a soft approximation in the optimization process, but how it is processed during downstream training is not stated in detail.
>
>
> **A2** We appreciate the reviewer’s insightful comment. To clarify, **our method uses the quantized synthetic data in low-bit form during downstream training**. Once the optimization is complete, **the final stored synthetic samples remain in their low-bit quantized form throughout downstream model training**.
>
>
> ---
>
> **W3** This paper claims that distilled datasets in low-bit formats can improve hardware utilization, but no empirical evidence or explanation is provided to support the claims.
>
>
>
> **A3** We sincerely thank the reviewer for raising this point. While our main focus is on condensation quality, our use of low-bit distilled data also naturally brings hardware-efficiency advantages. Below we clarify why such gains arise in practice.
>
> First, quantized inputs substantially reduce data-transfer and memory-bandwidth costs. Training often becomes bottlenecked not by computation but by moving data from CPU to GPU. Lower-bit synthetic images shrink the volume of data that must be loaded every iteration, improving cache locality and reducing I/O stalls. For instance, CIFAR-10 in FP32 requires roughly four times the transfer bandwidth of its INT8 counterpart at the same batch size. This advantage compounds during multi-epoch training, where the same distilled images are reused repeatedly.
>
> Second, low-bit distilled data aligns naturally with modern low-precision pipelines (e.g., PyTorch AMP, NVIDIA Tensor Cores). When inputs are already in INT8, the system avoids repeated FP32→INT8 casting at runtime, removing persistent overhead that accumulates over many training iterations and avoids performance degradation with our dataset fine-tuning strategy. Because quantized data occupies less GPU memory, it also enables larger effective batch sizes, improving device utilization and throughput. In practice, an 8-bit dataset can support larger batches than FP32 under the same memory budget, allowing more parallelism and more efficient kernel execution.
>
> ---

---

> ### Author Response · Authors · 2025-11-21
> **Authors Rebuttal (2/3)**
>
> **W4** This paper only compares the performance with full-precision dataset distillation methods, neglecting distilled data parameterization methods. Distilled data parameterization methods store the distilled data in a low-dimensional or implicit form, which can also reduce storage costs. This weakens the fairness of the evaluation and makes it difficult to demonstrate the effectiveness of the proposed method.
>
>
> **A4** Thank you for pointing out this! We would like to clarify that in Table 4 and Table 6 in the main text, we have compared to several data-parameterization based methods in our previouly submitted version. In the following, we compare the methods on (1) GAN-based parameterization, (2) Diffusion model-based parameterization, (3) Data Parameterization/factorization
>
> * **(1)** We also compare to additional **GAN-based parameterization**, H-PD [2] and GLaD [4]  by following the experiment setup in [2] in the following table.
>
> | Alg.  | Method | Tiny-ImageNet IPC-1 | Tiny-ImageNet IPC-10 | ImageNet-1K IPC-1 | ImageNet-1K IPC-10 |
> |-------|--------|---------------------|------------------------|--------------------|----------------------|
> |  | Pixel  | 2.6±0.1             | 16.1±0.2               | 0.1±0.1            | 21.3±0.6             |
> | SR²L       | GLaD   | 3.1±0.3             | 15.7±0.2               | 1.2±0.1            | 21.9±0.8             |
> |       | H-PD | 4.5±1.0         | 18.3±0.5           | 2.6±0.2        | 23.5±0.4         |
> |       | **BCDC (Ours)** |  **5.7±0.9**     |   **19.6±0.6**         | **3.7±0.3**       |     **25.1±0.6**    |
>
>
>
> * **(2)** **Diffusion model-based parameterization**: D$^3$HR [1] and D$^4$M [3] in Table 6 in the main text. We also provide results in the following table.
>
> | **Dataset**     | **Img/Cls** | **D⁴M** | **RDED**            | **CMI**             | **DWA**            | **+BCDC**          | **D³HR**           | **+BCDC (Ours)**          |
> |-----------------|-------------|---------|----------------------|----------------------|---------------------|---------------------|---------------------|---------------------|
> | Tiny-ImageNet   | 50          | 46.2    | 58.2 ± 0.1           | 53.7 ± 0.3           | 52.8 ± 0.2          | 55.1 ± 0.3          | 56.9 ± 0.2          | **59.3 ± 0.4**          |
> | Tiny-ImageNet   | 100         | 51.4    | ---                  | 56.9 ± 0.3           | 56.0 ± 0.2          | 59.6 ± 0.3          | 59.3 ± 0.1          | **61.8 ± 0.2**          |
> | ImageNet-1K     | 10          | 27.9    | 42.0 ± 0.1           | 38.5 ± 0.3           | 37.9 ± 0.2          | 39.6 ± 0.3          | 44.3 ± 0.3          | **46.9 ± 0.5**          |
> | ImageNet-1K     | 50          | 55.2    | 56.5 ± 0.1           | 55.6 ± 0.3           | 55.2 ± 0.2          | 57.5 ± 0.3          | 59.4 ± 0.1          | **62.6 ± 0.3**          |
> | ImageNet-1K     | 100         | 59.3    | ---                  | 59.8 ± 0.4           | 59.2 ± 0.3          | 61.7 ± 0.4          | 62.5 ± 0.0          | **64.7 ± 0.2**          |
>
> * **(3)** **Data Parameterization/factorization**: IDC [7], RememberThePast [6] and HaBa [5] in Table 4 in the main text. We also provide results in the following table.
>
>
>
>
> | **Dataset** | **Method**              | **MTT** | **IDC-I** | **IDC** | **HaBa** | **RememberThePast** |
> |-------------|--------------------------|---------|-----------|---------|----------|----------------------|
> | **CIFAR10 (Img/Cls = 1)** | Baseline | 46.3% | 36.7% | 50.6% | 48.3% | 66.4% |
> | | + BCDC (Ours) | **59.0%** | **50.1%** | **56.7%** | **65.3%** | **68.4%** |
> | **CIFAR100 (Img/Cls = 1)** | Baseline | 24.3% | 16.6% | 24.9% | 33.4% | 34.0%|
> | | + BCDC (Ours) | **31.2%** | **27.6%** | **33.4%** | **36.5%** | **36.2%** |
>
>
>
> Reference:
>
> [1] Taming diffusion for dataset distillation with high representativeness, ICML 2025
>
> [2] Hierarchical Features Matter: A Deep Exploration of Progressive
> Parameterization Method for Dataset Distillation, CVPR 2025
>
> [3] D^4M: Dataset Distillation via Disentangled Diffusion Model, CVPR 2024
>
> [4] Generalizing Dataset Distillation via Deep Generative Prior, CVPR 2023
>
> [5] Dataset distillation via factorization, NeurIPS 2022
>
> [6] Remember the past: Distilling datasets into addressable
> memories for neural networks, NeurIPS 2022
>
> [7] Dataset condensation via efficient synthetic data parameterization, ICML 2022

---

> ### Author Response · Authors · 2025-11-21
> **Authors Rebuttal (3/3)**
>
> **Q1** Given the same downstream training time budget, how do storage and downstream performance compared with the full-precision method?
>
> **A1**  Thank you for your question. We clarify the comparison under two evaluation settings used in our paper: (1) Same Number of Images (SI) and (2) Same Memory Budget (SM).
>
> * (1) Same Number of Images (SI).
> In this setting, we keep the number of synthetic images identical to the full-precision baselines. Since our synthetic data are stored in low-bit form, the resulting storage cost is substantially smaller than the full-precision method (achieving up to an 87.5% reduction in memory consumption). Importantly, the downstream training pipeline uses the same training-time budget. As shown in Table 3 of the main paper, our downstream accuracy remains comparable to the full-precision baselines in this setting.
>
> * (2) Same Memory (SM).
> We allocate the same total storage as the full-precision baseline. With quantized storage, we can therefore store more synthetic samples under the same memory budget. We then ensure that the downstream training uses a **same computation (SC)** budget as the baselines. Our method achieves significant performance improvements on TinyImageNet, as in the following table. The gain arises from the increased sample diversity enabled by quantized storage.
>
> | Img/Cls | DC      | +BCDC (SI) | +BCDC (SM&SC) | DSA     | +BCDC (SI) | +BCDC (SM&SC) | DM      | +BCDC (SI) | +BCDC (SM&SC) |
> |---------|----------|-------------|-------------|----------|-------------|-------------|----------|-------------|-------------|
> | 1       | 4.61±0.2 | 4.32±0.3    | **5.91±0.3** | 4.79±0.2 | 4.52±0.4    | **8.60±0.6** | 3.9±0.2 | 3.7±0.3    | **8.1±0.5** |
> | 10      | 11.6±0.3 | 10.2±0.3    | **16.12±0.4**| 14.7±0.2 | 13.61±0.5    | **18.23±0.5** | 12.9±0.4 | 11.6±0.4    | **16.5±0.6** |
>
> **Q2** It would be helpful if the authors could provide implementation details for downstream training and how the proposed method improves hardware utilization.
>
> **A2**  Thank you for your suggestions!
> * Implementation details: The downstream training procedure follows the same protocol as existing dataset distillation and dataset condensation baselines. Once our optimization pipeline produces the quantized synthetic dataset, no additional modifications to the downstream model or training loop are required. Thus, the downstream training protocol remains fully consistent with prior work, and our method integrates seamlessly with existing pipelines while providing significant storage and memory savings.
> * Hardware utilization, we would like to invite you to refer to **W3** and **A3**
>
> **Q3** The authors could consider briefly discussing how their proposed method differs from data parameterization dataset distillation methods based on latent variables, and may include them in a baseline comparison.
>
> **A3** Thank you for the suggestion.
>
> * Latent-variable or data-parameterization distillation methods operate by encoding training data into low-dimensional latent vectors and using a generator or decoder to reconstruct images. These approaches focus on *representation learning* and rely on auxiliary networks, generative priors, or learned embeddings. In contrast, our method works **entirely in pixel space** and introduces quantization-aware optimization to ensure that synthetic samples remain effective even when stored and used in *low-bit formats*. This is a fundamentally different design goal: latent-space distillation reduces dimensionality, whereas our framework explicitly targets *bit-level efficiency* to address bandwidth, memory, and hardware-precision constraints.
>
> * latent-variable methods typically assume *full-precision* latent vectors, decoders, and reconstructed outputs, meaning they do not reduce per-sample bitwidth. Our framework directly integrates quantization into the condensation process and optimizes synthetic data *under resource constraints*, solving a class of bottlenecks that latent-variable distillation does not address. Thus, the two lines of work are complementary, and our contribution fills a gap not covered by latent-space parameterization approaches.
>
> * For baseline comparisons, we would like to invite you to refer to **W4** and **A4**
>
> **Q4** Can this method be used for the distilled data generated from other tasks?
>
> **A4** Thank you for your suggestions. We perform experiments on reinforcement learning offline behavior distillation following the setup in [1] to distill state and action pairs. The results are shown in the following table:
>
>
> | Method | M-R | M   | M-E | M-R | M   | M-E | M-R | M   | M-E | Average |
> | ------ | --- | --- | --- | --- | --- | --- | --- | --- | --- | ------- |
> |    Av-PBC    |  35.9   |   36.9  |  22.0   |  40.9   |  32.5   | 38.7    |   55.0  |  39.5   |    42.1  |  38.2       |
> | Av-PBC+BCDC  | 38.0 | 41.7 | 26.6 | 43.0 | 34.6 | 42.5 | 57.2 | 43.9 | 45.0 | 43.1 |
>
>
> Reference:
>
> [1] Offline Behavior Distillation, NeurIPS 2024

---

### Author Response · Authors · 2025-12-03
**Rebuttal Summary**

We thank all the reviewers for their time and constructive feedback, as well as for their positive assessment of our contributions.

---

## strengths

**1. Novelty of the Idea** Reviewers **M8e6, gViH, 2oeQ** noted that the core idea—quantizing synthetic (condensed) data to low-bit precision is original. They emphasized that prior dataset condensation methods typically assume full-precision storage, so this work fills an important gap.



**2. Clear Motivation and High-Impact Problem** Reviewers **uiJF, 2oeQ** noted that BCDC is well-motivated and effectively enables the use of more synthetic samples, leading to performance that surpasses state-of-the-art baselines.



**3. Practical Significance** Reviewers **2oeQ, gViH** found the motivation *practical* and *addressing a key bottleneck of DC methods*, noting that reducing the storage footprint of distilled data improves deployability in *resource-constrained environments*.


**4. Theoretical Contributions** Reviewers **uiJF, gViH,  2oeQ** praised the *theoretical analysis*, specifically: analyze the trade-off between *quantization error and generalization* under memory limits and discuss on *Fisher information preservation*. They found the theory to add meaningful value beyond empirical evaluations.



**5. Strong Empirical Performance** Reviewer **M8e6** highlighted that *BCDC improves downstream accuracy*, enabling more synthetic samples and outperforming state-of-the-art baselines.


**6. Generality of the Framework** Reviewer **2oeQ** appreciated that our method is *general and compatible with existing condensation approaches*.



**7. Clear Presentations** Reviewer **uiJF** finds the paper well-written and appreciates the clarity of its presentation.



---




## Concerns

We also try our best to address the reviewers' concerns.

**comparison with distilled data parameterization methods**: We provide detailed comparisons with various state-of-art GAN-based parameterization, Diffusion model-based parameterization, factorization-based parameterization, etc, dataset distillation method.

**Hyperparameter selection:** We provide detailed hyperparameter selection and analysis of $k$ and $\lambda$

**generalization to other tasks** We provide reinforcement learning experiments dataset distillation experiments to show the genarlization of the proposed approach.

**computation cost on higher-resolution images** We provide runtime analysis on high-resolution ImageNet dataset.

**trade-off of per-sample fidelity favorable** We clarified that data distillation does not aim to preserve the content of original images. The synthetic images are meant to look different from the real ones, which, as shown in many prior studies, also helps protect data privacy.

---

### Meta-Review · Area_Chair_Apzu · 2025-12-24

**Summary:**

The most significant concerns shared by multiple reviewers are the computational cost and comparison with previous parameterization methods. Specifically, when the same memory allows for more quantized samples, it also brings more training consumption, which conflicts with the efficiency target of dataset distillation. On the other hand, the proposed method is an application of bit-conscious optimization framework to dataset distillation. While the authors claim that the method can be combined with many recent methods, there lack details of such integration.

Other concerns include clarification of low-bit training, hyperparameter analysis, etc.

**Reviewer Concerns:**

The authors well addressed the minor concerns. However, for the major concerns:
1. The authors didn't provide a transparent speed comparison between the proposed method and other non-parameterized methods. For 4-bit quantization, the same memory allow for 8-times the images of the standard precision. It brings significantly larger training effort, which is not marginal compared with full-set training. Also, the provided distillation time comparison with SRe2L is not fair. Note that the proposed method needs to generate many more images than the original SRe2L. The distillation also demands significantly more time.
2. While the authors provided the results of integrating the proposed quantization into more recent methods, such as DWA and D3HR, the details of such integration was not revealed. The framework of D3HR is largely different from bi-level optimization or surrogate-matching methods. Is the quantization directly applied to the generated images? The authors are encouraged to reveal more training details in the manuscript.

**Reviewer Scores:**

Based on that the major concerns haven't been addressed in this version, I think the reviewers would not change their score.

---

### Decision · Program_Chairs · 2026-01-26

Reject